# The negative regulator SMAX1 controls mycorrhizal symbiosis and strigolactone biosynthesis in rice

Jeongmin Choi[1✉], Tak Lee [2], Jungnam Cho [2,6], Emily K. Servante[1], Boas Pucker [1,7], William Summers [1], Sarah Bowden [3], Mehran Rahimi[4], Kyungsook An[5], Gynheung An[5], Harro J. Bouwmeester [4], Emma J. Wallington [3], Giles Oldroyd [1,2] & Uta. Paszkowski [1✉]

Most plants associate with beneficial arbuscular mycorrhizal (AM) fungi that facilitate soil nutrient acquisition. Prior to contact, partner recognition triggers reciprocal genetic remodelling to enable colonisation. The plant Dwarf14-Like (D14L) receptor conditions pre-symbiotic perception of AM fungi, and also detects the smoke constituent karrikin. D14L-dependent signalling mechanisms, underpinning AM symbiosis are unknown. Here, we present the identification of a negative regulator from rice, which operates downstream of the D14L receptor, corresponding to the homologue of the *Arabidopsis thaliana* Suppressor of MAX2-1 (AtSMAX1) that functions in karrikin signalling. We demonstrate that rice SMAX1 is a suppressor of AM symbiosis, negatively regulating fungal colonisation and transcription of crucial signalling components and conserved symbiosis genes. Similarly, rice SMAX1 negatively controls strigolactone biosynthesis, demonstrating an unexpected crosstalk between the strigolactone and karrikin signalling pathways. We conclude that removal of SMAX1, resulting from D14L signalling activation, de-represses essential symbiotic programmes and increases strigolactone hormone production.

[1] Department of Plant Sciences, University of Cambridge, Cambridge CB2 3EA, UK. [2] Sainsbury Laboratory, University of Cambridge, Bateman Street, Cambridge CB2 1LR, UK. [3] The John Bingham Laboratory, NIAB, Huntingdon Road, Cambridge CB3 0LE, UK. [4] Swammerdam Institute for Life Sciences, University of Amsterdam, Science Park 904, 1098 XH Amsterdam, The Netherlands. [5] Crop Biotech Institute, Kyung Hee University, Yongjin-si 446-701, South Korea. [6] Present address: CAS-JIC Centre of Excellence for Plant and Microbial Science, 200032 Shanghai, China. [7] Present address: Center for Biotechnology, Bielefeld University, Sequenz 1, 33615 Bielefeld, Germany. ✉email: jc913@cam.ac.uk; up220@cam.ac.uk

Arbuscular mycorrhizal (AM) symbioses are prevalent mutualistic interactions that involve most land plant species and fungi of the Glomeromycotina. Mutualism is achieved through a bi-directional exchange of nutrients where the plant host provides photosynthates to the obligate biotrophic fungus, which in return delivers soil minerals, primarily phosphate to the plant (for recent reviews see refs. [1,2]). At individual infection sites, development of the interaction follows a canonical series of steps. It commences with mutual recognition between the two partners in the rhizosphere, the first committed step in the interaction, followed by physical contact and tissue invasion. Root colonisation culminates in the formation of intracellular formation of tree-like fungal feeding structures, the arbuscules, which are the site of nutrient transfer.

The initial reciprocal recognition in the soil involves the release of the plant hormone strigolactone (SL) that activates fungal metabolism[3] and branching[4]. Known fungal signals on the other hand include chitooligomers (COs) and lipochitooligosaccharides (LCOs)[5,6]. Whereas the fungal SL receptors are unknown, plasma membrane-bound plant receptor-like kinases (RLKs) with chitin-binding Lysin Motifs (LysM) in their extracellular domain mediate recognition of the chitinaceous signals. These include legume Nod-Factor Receptors (NFR)1 and NFR for rhizobial nod-factors and Chitin Elicitor Receptor Kinase (CERK1) for chitin-based Pathogen Associated Molecular Patterns (PAMPs)[7]. The importance of these LysM RLKs for AM symbiosis was revealed in gene knock-down or knockout material of a variety of plant species, which displayed lower fungal colonisation[8–14]. The quantitatively reduced colonisation suggests that genetic redundancy might account for the absence of stronger AM phenotypes.

Perception of AM fungi by LysM RLKs triggers a calcium spiking response that leads to the activation of the common symbiosis signalling pathway (CSSP), which in legumes is equally required for the interaction with nitrogen-fixing rhizobia[15]. In AM symbiosis, CSSP activation causes extensive transcriptional reprogramming of the root to support the intraradical and intracellular accommodation of the fungus[1,2,16]. Mutation of CSSP signalling components in phylogenetically distant plant species consistently leads to severe impairment of fungal invasion of the root cortex cell layer[17,18], documenting the central importance of the CSSP for AM symbiosis development.

Indispensable for the perception of AM fungi in the rhizosphere is the α/β-fold hydrolase Dwarf14-Like (D14L) as evidenced by rice *d14l* mutants, which no longer responded to AM fungi with transcriptional changes[19]. The D14L homologue in *Arabidopsis thaliana* (KArrikin Insensitive, AtKAI2) is known to bind karrikins[20], smoke-derived butenolides that are involved in the restoration of vegetation by promoting seed germination and seedling growth after wildfire[21,22]. Interestingly, D14L is the evolutionarily older paralogue of the SL receptor Dwarf14 (D14)[23,24] pointing towards an ancient role of D14L in the perception of SL and karrikin related butenolides. In Arabidopsis, signal transduction of karrikin perception operates through the action of the E3 ligase *More AXillary Growth2 (AtMAX2)*[25] and the negative regulator *Suppressor of MAX2-1 (AtSMAX1)*[26]. The rice homologue of *MAX2, Dwarf3 (D3)*, is necessary for establishment of AM symbiosis as reflected by the absence of fungal colonisation of *d3* rice mutants which reproduced the *d14l* symbiosis phenotype[19,27]. Signalling components of the D14L/D3 signalling pathway that enable AM symbiosis are currently unknown.

Here we show that rice SMAX1 operates downstream of the D14L/D3 receptor, and negatively regulates AM symbiosis as reflected by the increased fungal colonisation and the suppression of the *d14l* and the *d3* mutant phenotypes in *smax1* double mutants. The absence of functional SMAX1 activates AM-induced genes in the absence of AM fungi, and induced SL biosynthesis. In addition, expression of a set of evolutionarily conserved genes, present in AM-host species, is dependent on functional *SMAX1*. Together, this suggests that the removal of the SMAX1 suppressor is required for transcriptional activation of the ancient AM symbiotic programme and for the control of SL production.

## Results

**Identification of rice *SMAX1*.** We hypothesised that analogous to Arabidopsis KAI2 signalling, a member of the rice *SMAX1-Like (SMXL)* gene family might be involved in the regulation of symbiosis downstream of D14L. A phylogenetic tree was built using the protein sequences encoded by *SMXL* family members of selected major model plant species (Supplementary Fig. 1). The SMXL family was divided into four subclades, corresponding to three known biological roles in karrikin signalling (SMAX1/SMXL2)[26,28], phloem development (AtSMXL3/4/5)[29], and in SL-signalling (AtSMXL6,7,8[30–32]; OsD53[33,34]). This is consistent with the prior phylogenetic analysis of the SMXL family in all plant lineages[35,36]. Within the SMAX1/SMXL2 subclade, the gene product of *LOC_Os08g15230* was found to be the closest homologue of AtSMAX1 (42% identity, 61% similarity). It was therefore termed OsSMAX1 and will be referred to as SMAX1 throughout the manuscript. Rice *SMAX1* consists of three exons, encoding a protein of 1040 amino acids (Supplementary Fig. 2). There are three predicted domains with a nuclear localisation signal (NLS) at the N-terminus, aligning to the functionally characterised NLS from homologous AtSMXL7[32], and a double-ClpN domain involved in protein interaction of chaperons and two P-loop ATPases.

The nuclear localisation signal suggested SMAX1 acts in the nucleus. Plasmids containing the coding sequences of *SMAX1, D14L* and *D3* were translationally fused with either the Green Fluorescent Protein (GFP) or Venus sequence and transiently expressed in rice leaf protoplasts under the control of the constitutive rice actin promoter. SMAX1 localised to the nucleus together with D3 and a proportion of the nucleo-cytoplasmically distributed D14L (Supplementary Fig. 3). The shared subcellular localisation of D14L, D3 and SMAX1 in the nucleus suggested a possible role in transcriptional regulation.

**Rice SMAX1 functions as a suppressor of AM colonisation.** To investigate the role of *SMAX1* during AM symbiosis, two T-DNA insertion mutants were obtained in the Dongjin cultivar from the public mutant collection[37,38]. Genotyping confirmed T-DNA insertions in the third exon (+2886 base pair (bp) from ATG) and in the 5'-untranslated region (−239 bp from ATG) in *smax1-1* and *smax1-2*, respectively (Supplementary Fig. 4a). Reverse transcription-(RT) PCR across the *SMAX1* insertion alleles revealed that in the *smax1-1* allele has a truncated mRNA, shortly before the insertion site (Supplementary Fig. 2 and 4b), whereas *smax1-2* produced an increased level of wild-type *SMAX1* mRNA, likely due to the action of 35 S enhancer sequences within the T-DNA construct (Supplementary Fig. 4c).

To examine the effect of mutated *SMAX1* on the AM symbiosis, both *smax1* mutants were inoculated with *Rhizophagus irregularis*. Quantification of fungal colonisation at an early stage 4-week post inoculation (wpi) revealed that the *smax1-1* mutant resulted in significantly higher total colonisation compared to wild-type ($p = 0.002$, Kruskal-Wallis test, $n = 4–5$), while *smax1-2* showed comparably low levels of colonisation to the wild-type ($p = 0.19$, Kruskal-Wallis test, $n = 4–5$) (Fig. 1a, b). Molecular characterisation of the extent of fungal root colonisation in both mutants and the wild-type revealed 10 times more abundant transcript levels of

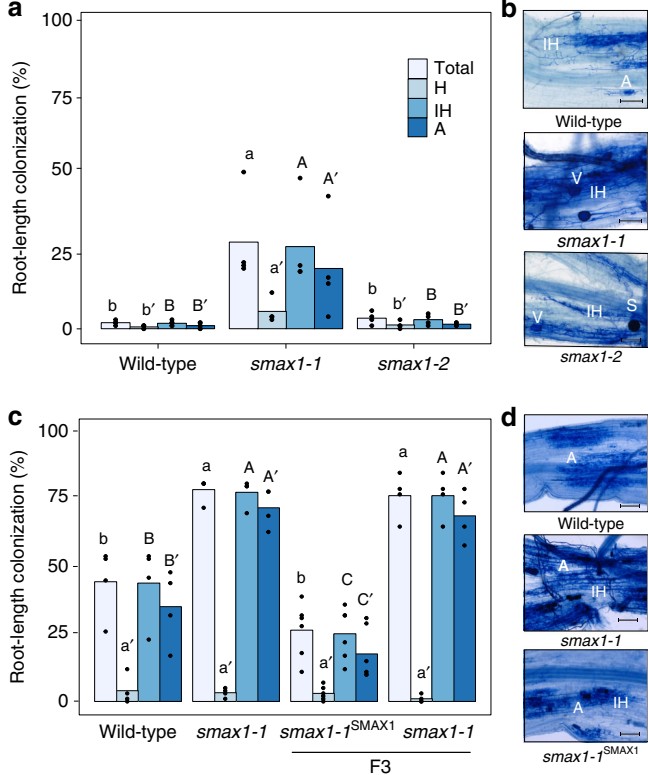

**Fig. 1 Root colonisation of *smax1* mutant and wild-type by *R. irregularis*.**
**a** AM colonisation levels of two *smax1* mutant alleles at 4 weeks post inoculation (wpi) with 300 spores. Each bar indicates the average percentage value of the respective fungal structure. Total, total colonisation, H, hyphopodia, IH, intraradical hyphae, A, arbuscules. Kruskal-Wallis test was performed, followed by post-hoc pairwise comparison using the Agricolae R package. Different letters represent significant difference (*p*-value < 0.05). Degrees of freedom = 2, wild-type (*n* = 5), *smax1-1* (*n* = 4), *smax1-2* (*n* = 4), Total: $\chi^2 = 8.60$, $p = 0.01$; H: $\chi^2 = 8.20$, $p = 0.02$; IH: $\chi^2 = 8.40$, $p = 0.02$; A: $\chi^2 = 8.32$, $p = 0.02$. **b** Micrographs of trypan blue stained roots for fungal structures in wild-type and *smax1* mutants at 4 wpi. Images represent roots from at least three independent plants of each genotype. IH intraradical hyphae, A arbuscule, V vesicle, S spore; scale, 100 μm. **c** AM colonisation levels of *smax1* mutant allele and the complementation line at 7 wpi. Degrees of freedom = 3, wild-type (*n* = 4), *smax1-1* (*n* = 4), *smax1-1^SMAX1^* (F3, *n* = 6), *smax1-1* (F3, *n* = 4), Total: $\chi^2 = 13.7$, $p = 0.003$; H: $\chi^2 = 3.01$, $p = 0.39$; IH: $\chi^2 = 13.9$, $p = 0.003$; A: $\chi^2 = 14.14$, $p = 0.003$. F3 indicates the segregating third generation plants after crossing *smax1-1* mutant with wild-type plants expressing the wild-type SMAX1 transgene. **d** Micrographs of trypan blue stained roots for fungal structures in wild-type, *smax1* and complementation line at 7 wpi.

the symbiosis marker *Phosphate Transporter (PT) 11*[39] mRNA in *smax1-1* relative to wild-type and the *smax1-2* allele (*p* = 0.01, Kruskal-Wallis test, *n* = 4–5, Supplementary Fig. 5a). Also at prolonged co-cultivation (7 wpi) there was a significant difference in total root colonisation of *smax1-1* (*p* = 0.023, Kruskal-Wallis test, *n* = 4–5) (Fig. 1c). Closer inspection of the fungal structures showed no alterations across the plant genotypes (Supplementary Fig. 5b). These data demonstrate that the increased *SMAX1* transcript levels in the *smax1-2* did not affect AM colonisation, whereas truncation of *SMAX1* in *smax1-1* resulted in an enhancement of fungal colonisation.

To verify that mutation of *SMAX1* caused increased fungal colonisation, we undertook genetic complementation of the *smax1-1* mutant with wild-type *SMAX1*. Homozygous *smax1-1* was crossed with a wild-type line, ectopically expressing *SMAX1*

under the control of the housekeeping rice *Actin1* promoter and the F₃ generation was genotyped for homozygous *smax1-1* individuals with segregating *pActin1::SMAX1* (*smax1-1^SMAX1^*). At 7 wpi, cosegregation analysis showed that *smax1-1^SMAX1^* plants reproduced the wild-type phenotype with significantly lower total colonisation relative to *smax1-1* (Fig. 1c, *p* = 0.0008, Kruskal-Wallis test, *n* = 4–6). Molecular phenotyping confirmed the microscopic results, with transcript levels of AM marker genes *AM1*, *AM3*, *PT11* and *AM14*[17] being significantly lower in the complemented *smax1-1^SMAX1^* lines as compared to *smax1-1* (*p* = 0.001, *p* = 0.0001, *p* = 0.0003 and *p* = 0.0008, respectively, Kruskal-Wallis test, *n* = 4–6) (Supplementary Fig. 6).

Together, these data inform that mutation of *SMAX1* results in increased colonisation by *R. irregularis*, suggesting that SMAX1 negatively regulates the AM symbiosis.

**SMAX1 functions downstream of the D14L/D3 receptor complex.** To define the epistatic relationship between SMAX1 and the receptor complex D14L/ D3, we sought to produce double homozygous lines between *d14l* or *d3* and *smax1*. Rice *D14L* was originally identified by genetic complementation of the *hebiba* mutant, harbouring a 170 kb deletion, containing *D14L*[19]. To generate genetically less complex material, the CRISPR/Cas9 technology was applied to create additional *D14L* loss of function alleles. Two single guide RNAs (sgRNAs) targeting the second exon of *D14L* produced small indels, thereby creating frameshift mutations leading to premature stop codons (Supplementary Fig. 7). Two alleles consisting of a GT (*d14l-1*) and a T deletion (*d14l-2*) resulted from the first sgRNA, while the third allele with a T insertion (*d14l-3*) was triggered by the second sgRNA. Next, we assessed the impact of the editing on gene function by examining AM colonisation. The three *d14l* alleles consistently showed strong defects in AM colonisation with below 5% colonisation whereas wild-type (cultivar Nipponbare) reached 88% (*p* = 0.045 for *d14l-1*, *p* = 0.025 for *d14l-2*, and *p* = 0.0008 for *d14l-3*, Kruskal-Wallis test, *n* = 3) at 7 wpi (Supplementary Fig. 8). Arbuscule development was not observed in any of the three alleles. This is consistent with the *hebiba* mutant phenotype, which also displayed no arbuscules as compared to high levels in corresponding wild-type. The microscopically determined phenotype was further supported by the molecular assessment of AM marker gene expression with no expression in the three *d14l* alleles equivalent to the *hebiba* deletion (Supplementary Fig. 9). The new *d14l* CRISPR alleles thus confirmed the essential role of *D14L* in AM symbiosis.

Double homozygous *d14l/smax1* and *d3/smax1* lines were generated by crossing *smax1-1* (referred to as *smax1* hereafter) with *d14l-2* (referred to as *d14l* hereafter) and *d3-1*[27] (referred to as *d3* hereafter), respectively (Fig. 2). AM colonisation was examined across the single and double mutant genotypes. As a control, siblings homozygous for *d14l* or *d3* but with functional *SMAX1* were also included. As expected *d14l* and *d3* single mutants displayed strong symbiosis defects with below 5% total fungal colonisation; however, *d14l/smax1* and *d3/smax1* double homozygous mutants had restored fungal colonisation with 84.8% and 79.6%, respectively (Fig. 2). By contrast, segregating siblings containing a wild-type *SMAX1* copy failed to rescue the *d14l* and *d3* mutant phenotype (Fig. 2). Consistently, AM marker gene expression correlated with fungal root colonisation, confirming the microscopic observations (Supplementary Figs. 10 and 11). In summary, the analysis of the epistatic relationship revealed that SMAX1 functions downstream of the receptor complex, D14L and D3, in the AM symbiosis.

To determine whether SMAX1 is regulated at the protein level, SMAX1 tagged with GFP was transiently expressed in leaf

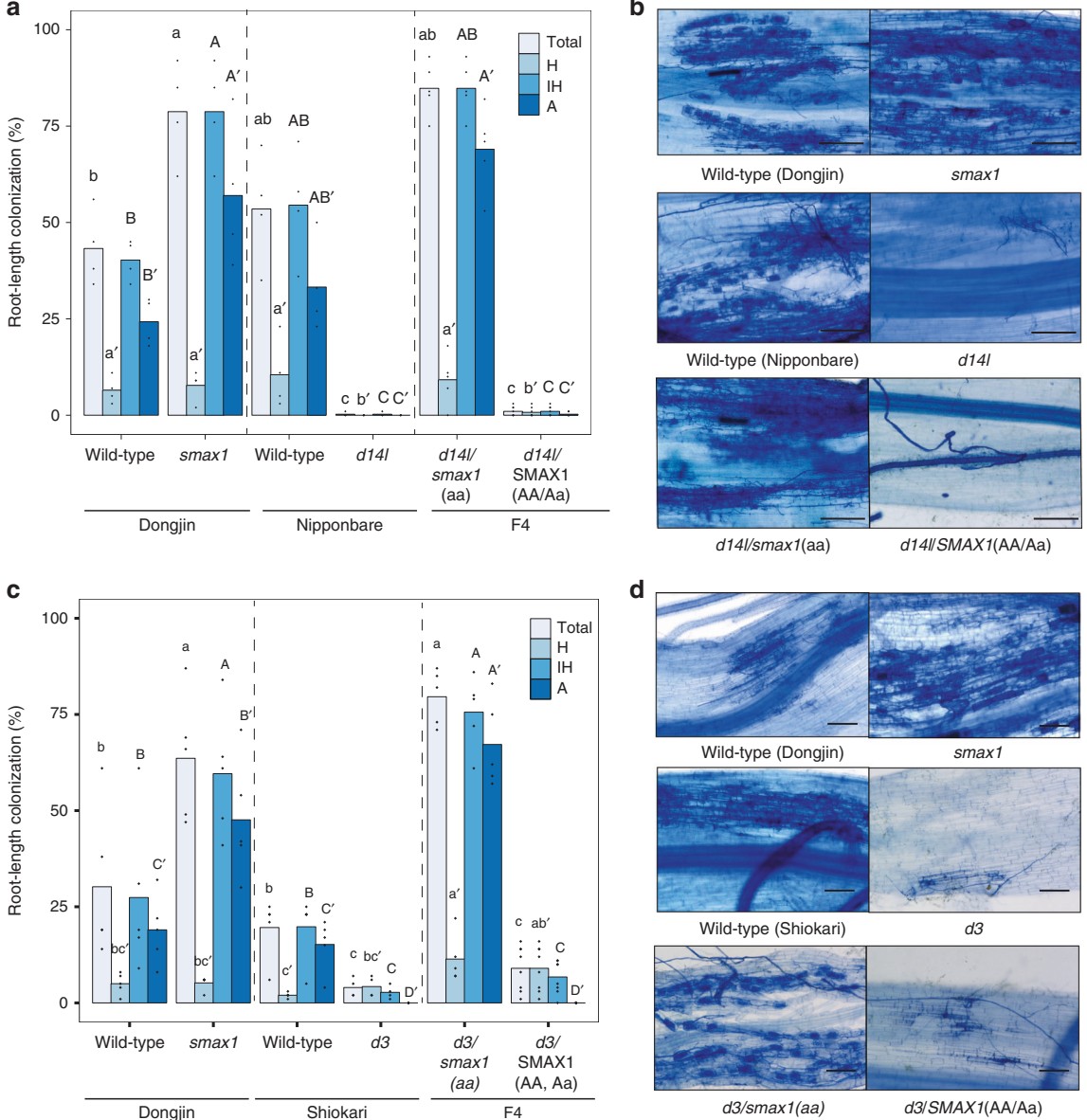

**Fig. 2 SMAX1 functions downstream of the karrikin receptor complex.** Root colonisation of mutant and wild-type rice genotypes by *R. irregularis*. **a** AM colonisation levels of a cross between *smax1* and the karrikin receptor mutant, *dwarf 14-like* (*d14l*), are quantified as a percentage (%) at 6 wpi with 300 spores. Dongjin and Nipponbare refer to the wild-type cultivars of respective mutant. F4 indicates the fourth generation progeny from the initial cross; *d14/smax1*(aa) indicates double homozygous alleles, whereas *d14SMAX1* (AA, Aa) refers to homozygous *d14l* with either wild-type or heterozygous *SMAX1* alleles. Each bar indicates the average percentage value of the respective fungal structure. Total total colonisation, H hyphopodia, IH intraradical hyphae, A arbuscules. Kruskal-Wallis test was performed, followed by post-hoc pairwise comparison using the Agricolae R package. Different letters represent significant difference (p-value <0.05). Degrees of freedom = 5, Dongjin wild-type (*n* = 4), *smax1* (*n* = 4), Nipponbare wild-type (*n* = 4), *d14l* (*n* = 4), *d14l/ smax1* (F4, *n* = 5), *d14l/SMAX1* (F4, *n* = 8), Total: $\chi^2 = 25.08$, $p = 0.0001$; H: $\chi^2 = 17.84$, $p = 0.003$; IH: $\chi^2 = 25.15$, $p = 0.0001$; A: $\chi^2 = 25.67$, $p = 0.0001$. **b** Micrographs of trypan blue stained roots for fungal structures in *d14l/smax1 and their parental lines*. Images represent roots from at least three independent plants of each genotype. Total total colonisation, H hyphopodia, IH intraradical hyphae, A arbuscules. Scale, 100 µm. **c** AM colonisation levels of crosses between *smax1* and *dwarf3* (*d3*). Dongjin and Shiokari refer to the wild-type cultivars of respective mutant. F4 indicates; *d3/smax1*(aa) indicates double homozygous alleles and *d3/smax1* (AA, Aa) refers to homozygous *d3* with either wild-type or heterozygous *SMAX1* alleles. Degrees of freedom = 5, Dongjin wild-type (*n* = 5), *smax1* (*n* = 5), Shiokari wild-type (*n* = 5), *d3* (*n* = 4), *d3/smax1* (F4, *n* = 5), *d3/SMAX1* (F4, *n* = 8), Total: $\chi^2 = 26.01$, $p = 0.0001$; H: $\chi^2 = 13.42$, $p = 0.02$; IH: $\chi^2 = 25.65$, $p = 0.0001$;A: $\chi^2 = 29.33$, $p = 2E0-5$. **d** Micrographs of trypan blue stained roots for fungal structures in *d3/smax1* and their parental lines.

protoplasts of wild-types, *d14l* and *d3* plants. We were unable to detect SMAX1-GFP expression in protoplasts from two different wild-type cultivars but could do so in protoplasts from *d14l* and *d3* mutants (Supplementary Fig. 12). This may reflect degradation of SMAX1-GFP due to D14L signalling in the presence of an unknown endogenous ligand, though further work would be

needed to confirm this. Notably, *SMAX1* transcript levels were significantly reduced in *d14l* and *d3* mutants compared to wild-type ($p = 0.00007$ and $p = 0.07$, respectively, Kruskal-Wallis test, $n = 4$–5) (Supplementary Figs. 10 and 11), suggesting the presence of a negative feedback mechanism, that may function in parallel with protein stability, to control SMAX1 protein

abundance. Also during SL-signalling, the SMAX1 homologue D53 (AtSMXL6/7/8) is degraded by the D14/D3 receptor complex upon SL perception[30–34]. Based on the detection of the SMAX1 protein in *d14l* and *d3* protoplasts (Supplementary Fig. 12), an equivalent mechanism might be involved here.

Collectively, our data suggest that presence of SMAX1 blocks initiation of the AM symbiosis in *d14l* and *d3* mutants, and that activation of the D14L-signalling pathway is required for the removal of SMAX1.

**Rice SMAX1 regulates mesocotyl elongation.** In Arabidopsis, the D14L-signalling pathway regulates plant development including hypocotyl elongation. Similarly, in rice, *d14l* and *d3* mutants have pronounced mesocotyl elongation compared to wild-type, demonstrating negative regulation by the D14L-signalling pathway[19,40]. To address functional conservation of SMAX1 in regulating epicotyl growth, we compared mesocotyl elongation from seedlings of *smax1, d3* and *d3/smax1* and the corresponding wild-type cultivars. We found that *smax1* developed significantly shorter mesocotyls than the wild-type ($p < 10E-5$, cultivar Dongjin, Kruskal-Wallis test, $n = 9–13$) (Supplementary Fig. 13), thereby indicating a conserved role of rice SMAX1 in suppressing mesocotyl elongation between Arabidopsis and rice. We next sought to establish the genetic relationship between *D3* and *SMAX1* for mesocotyl development. It was confirmed that *d3* had significantly longer mesocotyls than the wild-type (p = 0.002, cultivar Shiokari, Kruskal-Wallis test, $n = 9–13$) (Supplementary Fig. 13). However, mutation of *SMAX1* in the *d3* mutant background restored the long mesocotyl length to a range even shorter than wild-type seedlings ($p = 0.0001$ relative to Dongjin, $p = 0.04$ relative to Shiokari, Kruskal-Wallis test, $n = 9–13$), thus confirming that SMAX1 operates downstream of D3 in regulating mesocotyl elongation.

**SMAX1 controls AM symbiosis programmes and SL biosynthesis.** We performed transcriptome analyses on non-inoculated control roots to identify the downstream effects of SMAX1 removal in comparison to wild-type, *d3* and *d3/smax1* mutant plants (PRJNA593368 in the Gene Expression Omnibus database) (Supplementary Data 2). We identified a total of 11,674 differentially expressed genes (DEG, FDR adjusted $p ≤ 0.05$, fold-change $≥ |2|$) in at least one pairwise comparison (Fig. 3a, b, Supplementary Data. 3a), reflecting the significant role of SMAX1 and D3 in root signalling processes. Separate examination of the *smax1* and the *d3* transcriptome revealed 1130 and 7733 DEGs compared to the corresponding wild-type cultivars. The much larger number of DEGs in *d3* relative to *smax1* possibly indicates the collective impact of D3 on both SL and D14L signalling.

To reveal the relationship between the transcriptional signatures across the genotypes, hierarchical clustering analysis was performed. The genotypes clustered into three subgroups, the two wild-type cultivars (Dongjin, Shiokari), the two *d3* mutants, *d3* and *d3/SMAX1*, and the single *smax1* and double *d3/smax1* mutants (Fig. 3b). A large proportion of the *smax1* transcriptome, 77.3% (873 genes, Supplementary Data 3b), was similarly regulated in the *d3/smax1* double mutant. In contrast, only 37.6% (2905 genes, Supplementary Data 3c) overlapped between the *d3* and the double mutant. The relatedness of the *smax1* and the *d3/smax1* transcriptome is consistent with SMAX1 acting as a suppressor of D3.

Additionally, for a number of genes, we noticed increased transcript abundance in *d3/smax1* double as compared to *d3* and *smax1* single mutants (Supplementary Table 3e). To our surprise, the specific symbiotic phosphate transporter *PT11* was

reproducibly induced in *d3/smax1* mutant only, which was validated in independent experiments (Supplementary Fig. 14).

To generate understanding about the biological processes regulated by SMAX1 and D3, we conducted Gene Ontology (GO) enrichment tests with genes up-regulated in *smax1, d3* and *d3/smax1*. (Fig. 3d, Supplementary Data. 4). Terms for seed dormancy, response to mannitol stimulus, response to water deprivation, and response to light were significantly enriched in a SMAX1- and D3- dependent fashion ($p < 0.05$), consistent with their known roles in seed germination, drought tolerance and photomorphogenesis (Fig. 3d)[26,41,42]. On the other hand, genes involved in root development and response to auxin were overrepresented in *d3* alone, lending support for a role of D3 in SL-signalling modulating root development[43].

Derepression of genetic programmes in response to SMAX1 removal can include both induced or suppressed transcriptional activity. We therefore separately dealt with genes upregulated (UP) or downregulated (DOWN) in *smax1* and/or *d3/smax1* double mutants. To this end, an expression pattern cluster analysis was performed, employing the R package Mfuzz[44]. A total of 475 genes were upregulated in either *smax1* or *d3/smax1* as compared to wild-type or *d3* (*smax1*-UP, Fig. 3c, Supplementary Data.5). Representative gene expression profiles were validated in independent experiments (Supplementary Fig. 14). This included *D14-Like2a (DLK2a)*, a marker gene for D14L signalling[22], which was 255- and 712- fold upregulated ($p = 9.20E-09$, $p = 1.40E-09$, One-way ANOVA, $n = 3$ for Experiment I) in *smax1* and *d3/smax1* compared to Dongjin wild-type, respectively (Supplementary Fig. 14a). Consistently, in the complemented *smax1^SMAX1* line, mRNA levels of *DLK2a* were significantly reduced relative to the *smax1* mutant (Supplementary Fig. 15). By contrast, gene expression of *DLK2a* was 81-fold decreased in *d14l* compared to wild-type ($p = 0.0012$, two-sided student's *t*-test, $n = 3$) (Supplementary Fig. 16).

GO term enrichment tests ($p < 0.05$) of *smax1*-UP identified genes belonging to the biosynthetic pathway for isoprenoids, carotenoids and the resulting hormones, SL, gibberellic acid (GA) and abscisic acid (ABA, Fig. 3d, Supplementary Data 6). As SL is known to activate AM fungi for symbiosis[3,4], we extracted the gene expression profiles for all available biosynthetic genes. The key catalytic enzyme leading into the methylerythritol phosphate (MEP) pathway, *1-deoxy-D-xylulose 5-phosphate synthase2 (DXS2)*[45] and the beta carotene isomerase involved in the first committed step of SL biosynthesis, *Dwarf 27 (D27)*, were uniformly upregulated in *smax1, d3* and *d3/smax1* as compared to wild-types (Fig. 3e, Supplementary Fig. 17). Next, the genes involved in production of the SL precursor, carlactone, such as *Carotenoid Cleavage Dioxygenase7 and 8 (CCD7 and CCD8)* showed the same pattern. Rice has five *More Axillary Growth (MAX1)* cytochrome P450s, subfamily *CYP711*, required for the formation of the rice SL orobanchol[46]. Among these, *CYP711A2 Os01g0700900*, involved in conversion from carlactone to carlactonoic acid followed by deoxyorobanchol[47], was upregulated in *smax1, d3* and *d3/smax1* (Fig. 3e and Supplementary Fig. 17). The GRAS transcription factor *Nodulation Signalling Pathway2 (NSP2)* regulates the induction of SL production under certain conditions[48,49], and was also highly induced in *smax1* and *d3/smax1*, mirroring the significant increase in transcript levels of the SL biosynthesis genes (Fig. 3e). By contrast, transcripts of both *D27* and *CYP711A2 (Os01g0700900)* decreased in *d14l* while *DXS2* was not regulated (Supplementary Fig. 16). Thus, our data document that the entire SL biosynthesis pathway was transcriptionally upregulated when SMAX1 was absent. To verify possible correlation with SL production, we measured root SL contents across the different genotypes (Supplementary Fig. 18). We were able to measure the canonical rice SL 4-deoxyorobanchol (4-DO)

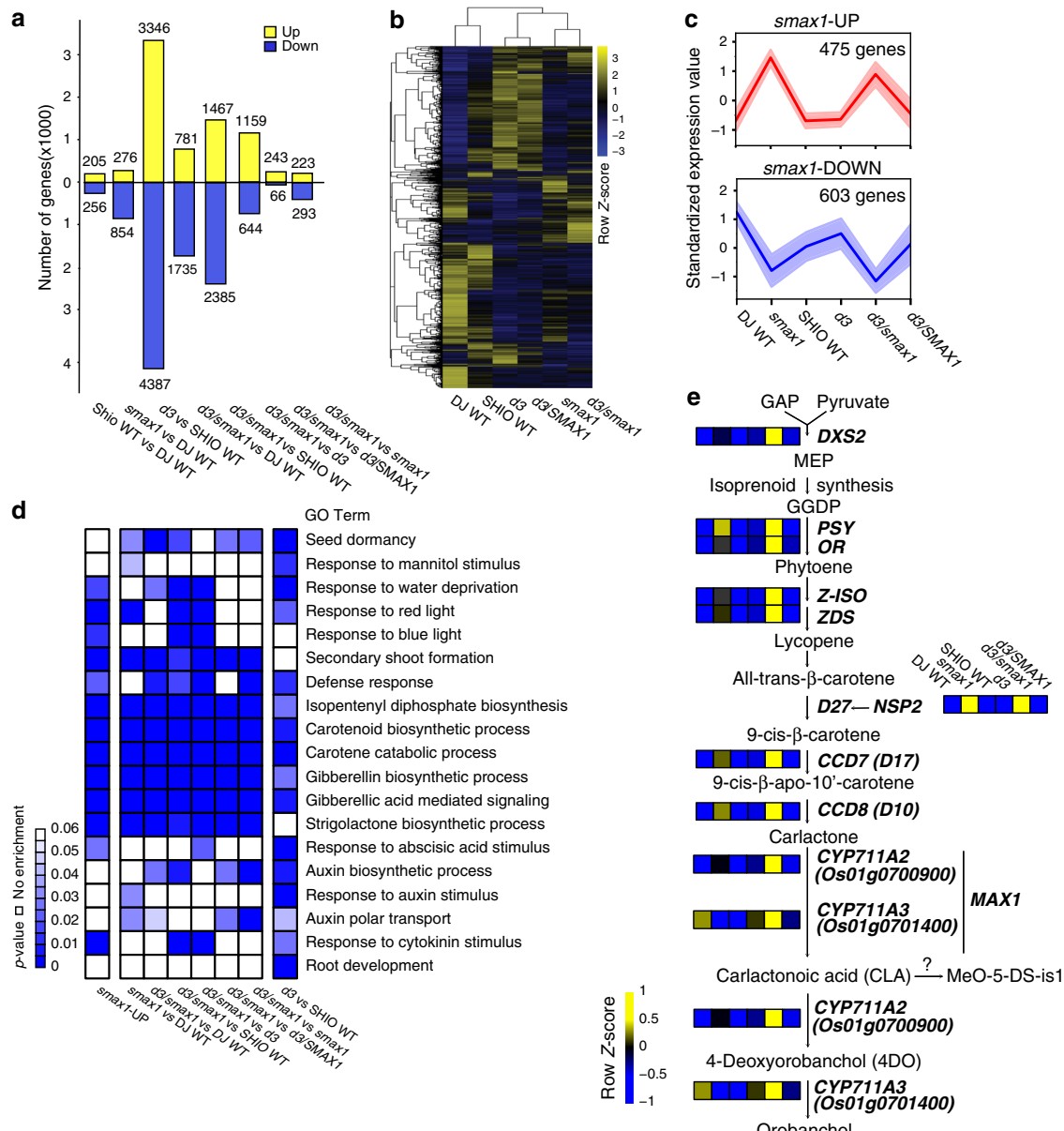

**Fig. 3 Transcriptome analysis of mock-inoculated mutant and wild-type rice genotypes. a** Overview over the number of differentially expressed genes ($p$-value $\leq 0.05$, fold-change $\geq |2|$) in pairwise comparisons as indicated. **b** Hierarchical clustering of 11,674 genes differentially expressed (FDR adjusted $p \leq 0.05$, fold-change $\geq |2|$) in at least one pairwise comparison was generated according to the Pearson correlation. The median expression values of genes from biological replicates were z-score normalized to have the mean of 0 and standard deviation (SD) of 1 across genotypes samples. (black: mean, blue: downregulated, yellow: upregulated). **c** Clustering of up (red)—and down (blue) -regulated genes in either *smax1* or *smax1/d3* by fuzzy c-means clustering. Graph shows a representative clustering pattern. Red and blue lines indicate the median expression value of all the genes in the cluster and the shadowed region indicates ±SD from median. The x and y-axis denote genotypes and the z-score normalized expression value. **d** Gene Ontology (GO) enrichment analysis of *smax1*- or *d3/smax1*-, upregulated genes from the pairwise comparisons. The enrichment rate is shown in $p$-value scale (<0.05), where lower values indicate stronger enrichment. **e** The metabolic pathway of the strigolactone biosynthesis pathway. Genes that are upregulated in either *smax1* or *smax1/d3* were selected and the normalized expression values across six genotypes were color coded according to z-score. *GAP glyceraldehyde 3-phosphate, DXS2 1-deoxyxylulose 5-phosphate synthase2, MEP 2-methyl-erythritol-4-phosphate, GGDP geranylgeranyl pyrophosphate, PSY phytoene desaturase, OR Orange-like protein, Z-ISO zeta-carotene isomerase, ZDS zeta-carotene desaturase, D27 Dwarf27, NSP2 nodulation-signaling pathway2, CCD7 carotenoid cleavage dioxygenase 7, D17 Dwarf 17, CCD8 carotenoid cleavage dioxygenase 8, D10 Dwarf 10, MAX1 More axillary growth 1, CYP711A2 and CYP711A3 cytochrome P450 monooxygenease, MeO-5-DS-is1 Me-5 methoxy-5-deoxystrigol isomer 1.*

in the wild-type Shiokari cultivar, and as a control, confirmed that 4-DO was not detected in the SL biosynthesis mutant *d27* (Shiokari cultivar). However, we were unable to detect 4-DO in the Dongjin cultivar, including *smax1*. The 4-DO levels in roots of the two SL perception mutants, *d3* and *d14*, were slightly elevated ($p = 0.08$, $0.13$, respectively, Kruskall-Wallis test,

$n = 3$-4). Interestingly, 4-DO was significantly increased ($p = 0.002$, Kruskal-Wallis test, $n = 3$-4) by seven-fold in the *d3/smax1* mutant, consistent with the collectively increased transcript levels of the SL biosynthetic genes. Another type of SL, methoxy-5-deoxystrigol isomer 1 (MeO-5-DS-is1)[50], was detected in *smax1* and *d3/smax1* confirming the increased generation of SL in the

absence of SMAX1. These data therefore suggest that D14L signalling positively regulates SL biosynthesis in rice roots by eliminating the negative regulator SMAX1.

A total of 603 genes were found to be significantly down-regulated in *smax1* or *smax1/d3* relative to wild-type or *d3* (*smax1*-DOWN, Fig. 3c, Supplementary Data. 7). GO enrichment analysis revealed that genes downregulated in *smax1* and *d3* shared similar categories across a diverse range of biological processes, such as response to karrikin and water transport, in line with the known role of the D14L-signalling pathway in karrikin perception and drought resistance (Supplementary Fig. 19, Supplementary Data 8 and 9)[22,42]. Collectively, in addition to providing further support for the previously described importance of D3 and SMAX1 in aspects of plant development, the genes whose transcription is regulated by SMAX1 identified a regulatory link between D14L-signalling and the biosynthesis of major carotenoid-derived hormones, in particular SL.

To identify from the *SMAX1*-regulated genes those of potential relevance for AM symbiosis, we compared the *smax1*-UP with genes that were previously reported to be upregulated during colonisation of rice roots by *R. irregularis*[51] (Fig. 4a). A large proportion of 135 genes (28.4%) with known induced expression profiles in response to fungal colonisation of wild-type roots displayed activated transcription in *smax1*-UP (Fig. 4a, categories I and III), indicating that the SMAX1 regulon exerts a degree of control over the expression of these AM-associated genes. Next, we compared *smax1*-UP with genes conserved in mycorrhizal but absent from non-mycorrhizal plant species[52]. We found that a significant fraction of the evolutionarily conserved genes (21%)[52] were transcriptionally regulated by the presence of SMAX1 (Fig. 4a, categories II and III, Supplementary Data. 10), consistent with the ancient nature of the D14L-signalling pathway components. Among these, transcripts of 9% (42 genes) of the *smax1*-UP genes transcription were not detected in the wild-type (read count 0) but accumulated to significant levels ($p < 0.05$) in both *smax1* or *d3/smax1* (Fig. 4a numbers in parenthesis, Supplementary Data 11). Together, transcription of these genes appeared to strictly depend on the absence of SMAX1, suggesting that their expression occurred upon activation of the D14L-signalling pathway.

Next, we examined several genes with known roles in AM symbiosis such as critical CSSP signalling components. The kinases *SYMRK* (Fig. 4a, category II)[53,54], *CCaMK* (Fig. 4a, category IV)[17,55,56] and the transcription factor *CYCLOPS* (Fig. 4a, category III)[17,57], are under the control of the SMAX1 regulon. In addition, and as mentioned earlier, the *smax1*-UP gene list showed an overrepresentation of GO terms associated with SL biosynthesis (Fig. 3d, Fig. 4a, Supplementary Data. 6). Furthermore, three kinases, *LysM-RLK2* (*NFR5*) required for perception of fungal LCOs[12–14,58], *LysM-RLK3* (*LYK1*) and *Kinase 6* (*KIN6*), the closest homolog of Arbuscular Receptor-like Kinase (*OsARK1, AM14*) associated with AM development[59], were also found on this list. Increased transcript levels of *NFR5*, *LYK1* and *KIN6* in *smax1*-UP were confirmed in independent experiments (Supplementary Fig. 14a).

Consistent with the presence of SMAX1 having a negative effect on transcriptional activators, a considerable number of transcription factors were part of the *smax1*-UP list (Fig. 4d). In addition, we observed that a total of 88 genes from the *smax1*-DOWN list were also suppressed in wild-type roots during mycorrhizal colonisation (Fig. 4b). GO enrichment analysis of the *smax1*-DOWN gene list identified overrepresentation of genes involved in cell-wall organisation (Supplementary Fig. 19). Of these, 17 genes showed commonly decreased transcript levels in wild-type roots when colonized by AM fungi[51]. The majority of these are predicted to encode expansins and peroxidases, implicated in secondary cell-wall modification[60]. In the context of AM symbiosis, it was hypothesized that reduction of such cell-wall modification genes could either support the AM-induced root architectural changes, or alternatively facilitate intercellular fungal proliferations[51].

In summary, we found significant overlap between the SMAX1 regulon and genes required for AM symbiosis or evolutionarily conserved across mycorrhizal plant species (Fig. 4c). Unexpectedly, genetic removal of *SMAX1* also led to the induction of SL biosynthesis. Together this lends support for SMAX1 as an essential regulator of AM symbiosis programmes and SL hormone balance.

## Discussion

Transcriptional derepression occurs commonly in plant hormonal signalling to enable rapid responses to environmental changes[61]. Here, we demonstrate that the AM symbiosis depends on double negative regulatory mechanisms that link hormonal and pre-symbiotic signalling. We identified rice SMAX1 as a suppressor of AM symbiosis development, functioning downstream of the D14L/D3 receptor complex since the *SMAX1* mutation fully restored AM fungal colonisation in the *d14l* and *d3* mutant backgrounds. We conclude that activation of the D14L-signalling pathway results in removal of SMAX1 (which is further supported by the detection of SMAX1 protein in *d14l* and *d3* mutant but not in wild-type) to de-repress critical symbiosis programmes. The significant changes in the transcriptome profile between *smax1* and wild-type however suggests a qualitative difference between removing SMAX1 genetically or through signalling turnover.

Guided by hormonal signalling pathways involving derepression, we hypothesised that SMAX1 directly or indirectly negatively regulates the activity of transcriptional regulators of AM symbiosis programmes (Fig. 5). Consistent with this, the transcriptome analysis of *smax1* single and *d3/smax1* double mutants revealed genes induced and repressed in *smax1*-UP and *smax1*-DOWN gene sets, suggesting that transcriptional activators and repressors might both be targets of SMAX1 suppression. Indeed, a large number of transcription factors themselves were transcriptionally regulated in the absence of functional SMAX1, including the CSSP members *CYCLOPS*[57], which is vital for AM symbiosis establishment, and *NSP2*, which regulates the induction of SL biosynthesis[48,49]. Genes across the entire SL synthesis pathway showed significantly increased transcript levels in *smax1* single and *d3/smax1* double mutant lines, and apparently resulted in the increased production of the SL hormone itself.

*CYCLOPS* and several other genes including CSSP signalling components (*NFR5, SYMRK, CCAMK*) also showed induction in *smax1* mutant backgrounds. The strong *d14l* symbiosis phenotype cannot easily be explained by modification of SL production alone[19] but might be attributable to the collective alteration of the transcriptional activity of several CSSP components. In summary, the data shown here are consistent with D14L signalling controlling the integration of SL biosynthesis and AM symbiosis development (Fig. 5).

Knowledge on the chemical nature of ligands that bind and activate D14L-signalling is currently restricted to karrikin, which in nature requires pyrolysis of plant cellulose or sugars (for review see ref. [62]). However, the developmental phenotypes of *A. thaliana kai2* argue for the production of endogenous karrikin-like binding substrates for D14L[63]. The insensitivity of *d14l* rice roots to AM fungi might be brought about by the need for D14L-signalling activation via either an endogenous or a fungal signal[19]. Our failure to observe detectable levels of SMAX1 protein in wild-type protoplasts is consistent with the availability of an endogenous ligand that, via binding to D14L would launch the signal transduction resulting in SMAX1 removal (Fig. 5). We therefore

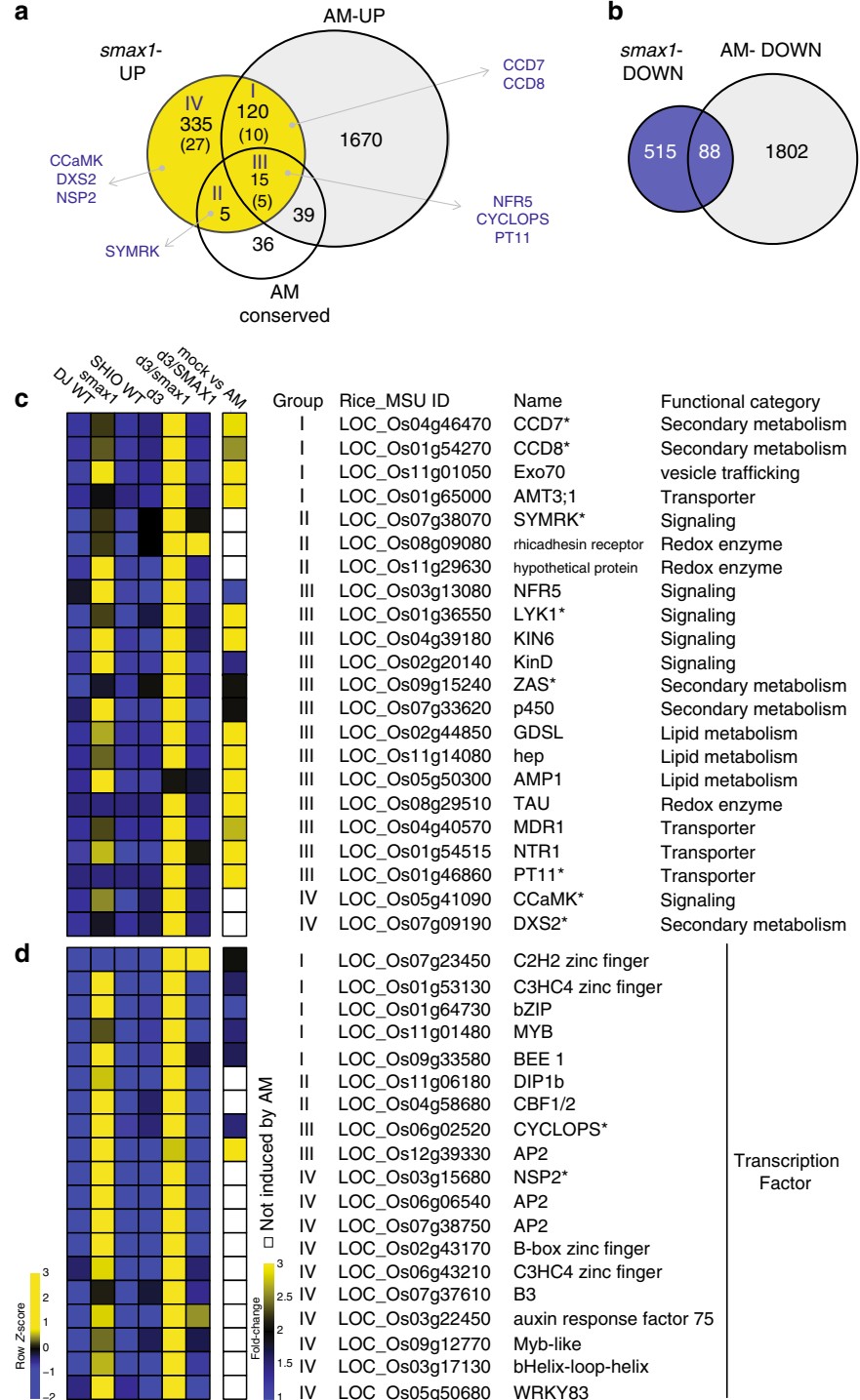

**Fig. 4 SMAX1 regulates AM-associated genes. a** The Venn diagram shows commonly upregulated genes in *smax1* and during AM symbiosis[51], or overlapping with genes conserved in AM-host plant species[52]. *smax1*-UP represents the genes upregulated either in *smax1* or *d3/smax1*. The number in parenthesis indicates the number of genes that are not expressed in wild-type but induced in either *smax1* or *d3/smax1*. Genes with known function in AM symbiosis are shown in capital letters. **b** The Venn diagram displays commonly suppressed genes in *smax1*-DOWN and during AM symbiosis[51]. **c, d** The heatmap depicts relative expression levels and fold-change of induction during AM symbiosis of key genes involved in AM symbiosis, and of AM conserved genes (**c**) and transcription factors (**d**) that were induced in either *smax1* or *d3/smax1*. Asterisk indicates genes whose functions are previously characterized (See Supplementary Table 2).

propose that an endogenous signal conditions the root for pre-symbiotic recognition of AM fungi via activation of the D14L-signalling pathway leading to elimination of SMAX1 suppression.

Noteworthy is also the frequently observed enhanced effect in the amplitude of the *smax1*-UP gene list in *d3/smax1* double

knockout, which was particularly pronounced for the genes encoding the SL biosynthesis pathway. It is possible that the negative feedback mechanism, which results in increased SL synthesis in the SL-signalling mutant *d3*[64] may account for the boosted transcript levels in the double relative to the single *d3* and

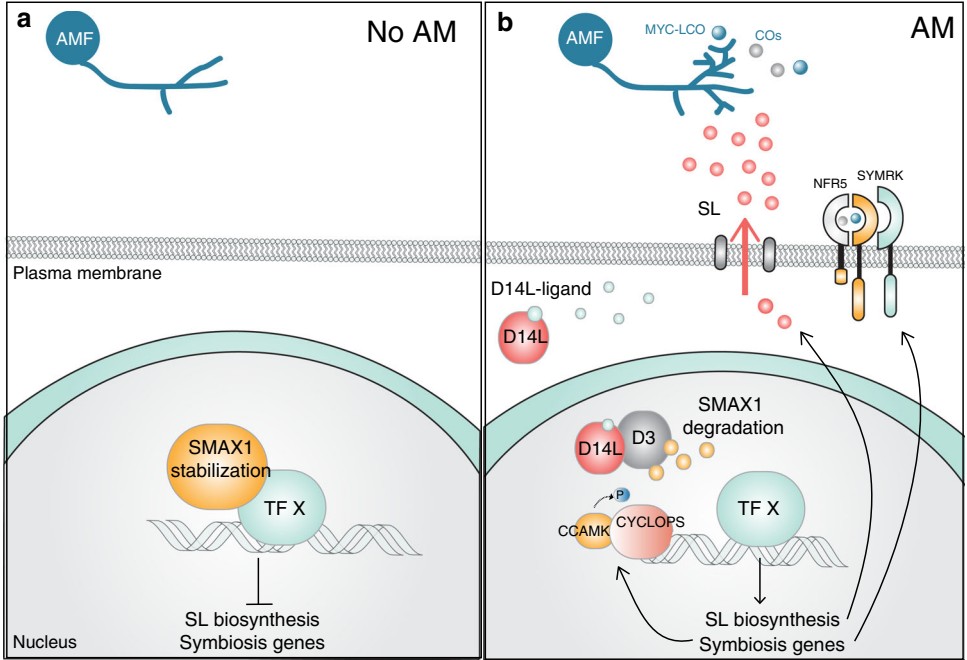

**Fig. 5 Hypothetical model for the action of SMAX1 during AM symbiosis.** The proposed model illustrates that SMAX1 is stabilised in *d14l* and *d3* mutants, resulting in severely defective AM colonisation, suppressing transcription of symbiosis programmes. By contrast, in wild-type an unidentified D14L ligand binds to D14L, activating D14L signalling and degradation of SMAX1, thereby de-repressing relevant transcription factor/s. The symbiosis and SL biosynthesis genes are de-repressed enabling AM symbiosis and SL production.

*smax1* mutants. However, the combinatorial effects extend also into SL-unrelated genes, surprisingly including the upregulation of the AM specific phosphate transporter *PT11* in *d3/smax1* but not in *smax1* backgrounds. The activation of AM symbiosis specific phosphate transporters in the absence of AM colonisation has been reported previously, and resulted from the over-expression of transcription factors such as *RAM1*[65], *CBX1*[66] and *WRI5a*[67]. While the mechanism underpinning the transcription of *PT11* in *d3/smax1* plants is unclear, it may be that either D3 or SMAX1 (or both) serve additional signalling functions, independent of D14 and D14L, which only in combination de-repress relevant transcription factors.

After the divergence of monocots and dicots, *SMAX1* duplication gave rise to a closest paralog, *SMXL2* (Supplementary Fig. 1). In Arabidopsis, SMXL2 functions redundantly to SMAX1 in plant development[28,68,69]. While it is possible that SMXL2 also plays a role in AM symbiosis, a loss of function of *SMAX1* alone recovered the lack of AM colonization in both *d14l* and *d3* mutants (Fig. 2). This suggests limited functional redundancy for AM symbiosis between SMAX1 and SMXL2.

In summary, AM symbiosis requires the degradation of SMAX1 to transcriptionally activate genes vital for communication and accommodation with the AM fungi and for regulating SL balance. In addition, the activation of the evolutionarily conserved AM genes through SMAX1 removal suggests an ancient regulatory role of the D14L-signalling pathway in AM symbiosis.

## Methods

**Plant materials**. All rice lines used in this study are *Oryza sativa* L. ssp. *japonica* varieties. Two *smax1* T-DNA insertion lines, *smax1-1* (3D-00402) and *smax1-2* (2D-00142), were generated in the Dongjin cultivar[37,38]. The *d3* mutant contains a putative transposable element insertion at the first exon[70] in the Shiokari cultivar. Strigolactone biosynthesis (*d27*) and signaling mutant (*d14*) are also in the Shiokari cultivar[70]. The *hebiba* mutant arose in the Nihonmasari cultivar and contains a 170 kb deletion as a result of fast neutron radiation mutagenesis. The *hebiba*^AOC^ was generated by introducing the jasmonate biosynthesis gene, *Allene Oxide Cyclase (AOC)* with the native promoter into the *hebiba* mutant background[19]. All *d14l* CRISPR lines were generated in Nipponbare cultivar as described below.

**Plant growth and conditions for AM colonisation**. Seeds were sterilized in 70% (v/v with water) ethanol briefly and incubated with 3% (v/v with water) sodium hypochlorite solution on a shaker at room temperature for 20 min. Seeds were rinsed with autoclaved water three times before planting on 0.6% (w/v with water) Bacto agar. For AM colonisation, seedlings were grown for four days at 30 °C in the dark and were transferred into cones containing sand and 300 *Rhizophagus irregularis* fungal spores (unless otherwise described). Fungal spores were freshly extracted from transgenic hairy carrot root cultures on the day of inoculation and were resuspended in water. The plants were grown in a walk-in growth chamber where day/night length is 12/12 h at 28/20 °C with 60% relative humidity. For the first week post inoculation, plants were given water to promote AM colonisation. From the following week plants were fertilized with half-strength Hoagland solution containing 25 µM of phosphate with 0.01% (w/v) iron supplement, Sequestrene Rapid (Syngenta) twice a week.

**Phylogeny tree construction**. Homologous sequences of *Arabidopsis thaliana* SMAX1 were retrieved from sources listed below. The final sequence set (Supplementary Data 01) was aligned via MAFFT v7.299b[71]. To exclude bias through the order of input sequences in this alignment step, the sequence order was shuffled 100 times and aligned again via MAFFT. Phyx v0.1 was applied to trim columns with low occupancy (-p 0.1) from the alignment[72]. The final tree was generated with RAxML v8.2.11[73] with the following parameters: '-f a -p 12345 -x 12345 -c 25 -m PROTGAMMAAUTO -# 10.

Source of sequences are as below.

*Medicago truncatula*, Medicago truncatula Genome Database, (http://www.medicagogenome.org[74]), 2019

*Lotus japonicus*, Lotus base, (https://lotus.au.dk[75]), 2019

*Oryza Sativa*, Rice genotype annotation project, (http://rice.plantbiology.msu.edu[76]), 2019

*Arabidopsis thaliana*, The Arabidopsis Information Resource, https://www.arabidopsis.org), 2019

*Brachypodium distachyon* v3.1, *Zea mays* Ensembl_18, *Hordeum vulgare* r1, *Marchantia polymorpha* v12.1 (https://phytozome.jgi.doe.gov/pz/portal.html[77], 2019.

**Fungal structure quantification and microscopic imaging**. Roots were stained with Trypan blue and mounted on a glass slide before observation under a brightfield microscope according to a modified gridline intersect method[78]. Representative images were captured by Keyence VHX-5000 Digital microscope (Keyence, Milton Keynes, UK). For the detailed arbuscule images, roots were stained with 5 µg/mL wheat germ agglutinin conjugated with Alexa 488 fluorescent dye in 1X phosphate-buffered saline solution (pH 7.4). After 7 days incubation at 4 °C in the dark, the roots were counterstained with 5 µg/mL of propidium iodide to visualize cell-wall. Images were taken by a Leica SP8 microscope with excitation

at 488 nm and 514 nm, emission at 524 nm and 617 nm for Alexa488 and propidium iodide, respectively.

**Rice transformation**. Callus for transformation of rice cultivars Nipponbare and Dongjin was generated by plating surface-sterilised mature seed, with embryo axes removed, on N6DT medium (3.95 g/L N6 basal salts, 30 g/L sucrose, 300 mg/L casein hydrolysate, 100 mg/L myo-inositol, 2878 mg/L proline, 0.5 mg/L nicotinic acid, 0.5 mg/L pyridoxine HCl, 1 mg/L thiamine HCl, 37.3 mg/L $Na_2EDTA$, 27.8 mg/L $FeSO_4$, 2 mg/L 2,4-D Na salt, 150 mg/L Timentin, 4 g/L Gelrite, pH 5.8). Plates were sealed with Parafilm and cultured in the dark at 28 °C for 17 days. Callus was cut into 2–4 mm pieces, plated on fresh N6DT and cultured as before for a further 4 days. Agrobacterium strain EHA105 was cultured at 28 °C overnight on YEP medium with appropriate antibiotic selection. Bacteria were scraped from the plate and resuspended in 1 mL AAM medium with 20ug/mL acetosyringone (Sigma Aldrich). Approximately 500µLs of this suspension was then transferred to a conical flask containing 50 mL AAM with acetosyringone (20ug/mL) and adjusted as necessary to give an $OD_{600}$ of 0.2–0.3. The Agrobacterium suspension was incubated at 28°C with shaking (150 rpm) for 1.5–2 h after which time 10% pluronic F-68 (10µL/mL, Sigma Aldrich) was added. Rice callus was transferred to Universal tubes, inoculated with 10 mLs of the Agrobacterium culture and incubated on the bench for 5 min with gentle agitation. Following removal of the inoculum the callus was transferred to a Petri dish containing sterile filter paper, sealed with Parafilm, and co-cultivated for 3 days at 25 °C day/23 °C night in the dark.

Callus pieces were transferred to N6DT medium containing 50 mg/L hygromycin, plates sealed with micropore tape (3 M) and cultured in the dark at 30 °C for 10 days. Callus was then transferred to RPT medium (4.3 g/L MS salts, 30 g/L sucrose, 300 mg/L casein hydrolysate, 100 mg/L myo-inositol, 500 mg/L proline, 500 mg/L glutamine, 0.5 mg/L nicotinic acid, 0.5 mg/L pyridoxine HCl, 0.1 mg/L thiamine HCl, 37.3 mg/L $Na_2EDTA$, 27.8 mg/L $FeSO_4$, 1 mg/L NAA, 2 mg/L BAP, 5 mg/L ABA, 150 mg/L Timentin, 5 g/L Gelrite) containing 50 mg/L hygromycin, again sealed with micropore tape and cultured in the dark for a further 5 days. Viable callus was then transferred to RMT medium (4.3 g/L MS salts, 30 g/L sucrose, 1000 mg/L casein hydrolysate, 100 mg/L myo-inositol, 0.5 mg/L nicotinic acid, 0.5 mg/L pyridoxine HCl, 0.1 mg/L thiamine HCl, 37.3 mg/L $Na_2EDTA$, 27.8 mg/L $FeSO_4$, 0.5 mg/L NAA, 3 mg/L BAP, 150 mg/L Timentin, 5 g/L Gelrite) containing 25 mg/L hygromycin (RMT25H), sealed with micropore tape and cultured at 28 °C (day)/26 °C (night) with a 16 h daylength (100 uE $m^{-2}s^{-1}$). After 10 days, regenerating material was transferred to fresh RMT25H. Regenerating shoots were transferred to OS3T medium (2.2 g/L MS medium, 15 g/L sucrose, 500 mg/L MES, 150 mg/L Timentin, 2 g/L Phytagel) containing 25 mg/L hygromycin (OS3T25H) in small pots (Greiner 175 ml lidded).

Leaves from rooted plantlets were sampled into 2 mL Safelock tubes containing two 3 mm steel balls and tissue ground using a Genogrinder set to 50 s/m for 30 seconds. Genomic DNA was extracted using sucrose extraction buffer (50 mM Tris-HCl pH 7.5, 300 mM NaCl, 300 mM sucrose), tubes incubated at 95–100 °C for 10 min in a heating block and micro-centrifuged for 30 seconds at 14,000 rpm. The supernatant was then used in multiplex PCR to confirm transformation using primers for the hpt selectable marker gene (Supplementary Table 3) and a cyclophillin2 endogenous control gene (Supplementary Table 3)[59].

**RNA extraction and quantitative RT-PCR**. Harvested roots were frozen in liquid nitrogen and stored at −80 °C until extraction. Roots were homogenized with two 4 mm metal bearings in 2 mL tubes by using TissueLyser II (Qiagen) at 30 Hz for 1 min twice. Total RNAs were extracted by Trizol method. The RNA purity and integrity were confirmed by PCR with GAPDH primers (Supplementary Table 3) spanning an intron[79] and RNA electrophoresis in 2.5% (w/v) agarose gel, respectively. One microgram of RNA was treated with DNase I prior to cDNA synthesis using reverse transcriptase II (Invitrogen). Quantitative RT-PCR was performed by measuring the intensity of SYBR Green Fluorescent dye conjugated to double-stranded DNA molecules using CFX96 Touch Real-Time PCR detection system (Bio-Rad). Gene expression values were normalized to geometric mean of three rice housekeeping genes, *Cyclophillin2, GAPDH* and *Ubiquitin*[11,80].

**Subcellular localisation**. *D14L*, *D3* and *SMAX1* coding sequences (CDs) were synthesized by GeneArt (Life Technologies). *D14L* and *D3* constructs were assembled according to the Multisite Gateway system (Invitrogen). *D14L* and *D3* CDs were cloned into the *pDONR221* vector using Gateway (Invitrogen) BP reaction. The rice *Actin* promoter containing an intron was amplified from GoldenGate Level 0 module (EC15216) and introduced into the *pDONR1R4* vector using Gateway reactions. The *Venus* coding region was cloned in the *pDONR2R3* and the final constructs were assembly into the *pB7m34GW* vector by LR reactions. *SMAX1* constructs were cloned according to GoldenGate Cloning system[81,82]. Protoplast transformation was conducted with 7 day-old leaf protoplasts grown in half-strength solid MS media (1% BactoAgar) at 30 °C in the dark[19]. Finely chopped leaves (1 mm) were incubated in the enzyme solution (1.5% Cellulase RS (Duchefa Biochemie), 0.75% Macrozyme R-10 (Duchefa Biochemie), 0.6 M mannitol, 10 mM MES buffer pH 5.7, 10 mM $CaCl_2$, and 0.1% BSA) for 5 h in the dark with gentle shaking (60 rpm, 28 °C). The digestion was stopped by adding equal

volume of W5 solution (154 mM NaCl, 125 mM $CaCl_2$, 5 mM KCl and 2 mM MES at pH 5.7). Protoplasts were collected by filtering through 40 µm nylon mesh followed by centrifugation at $300 \times g$ for 4 min at 4 °C. Cells were washed with W5 solution twice and resuspended in MMG solution (0.4 M mannitol, 15 mM $MgCl_2$ and 4 mM MES at pH 5.7). For transfection, 5 µg of freshly prepared plasmids were incubated with $4 \times 10^5$ cells of protoplasts in the presence of equal volume of PEG solution (40% (W/V) PEG 4000; Fluka, 0.2 M mannitol and 0.1 M CaCl2) for 15 min in the dark. The transfection was terminated by adding the same volume of W5 solution. Finally, the supernatant was removed by gentle centrifugation ($300 \times g$ for 4 min at 4 °C) and the protoplast were resuspended in WI solution (0.5 M mannitol, 20 mM KCl and 4 mM MES at pH 5.7) for 12 h in the dark. The images were taken using confocal laser scanning microscope Leica TCS SP5. GFP and Venus fluorescence were excited with argon laser at 488 nm and 514 nm and captured with a 490–550 nm and 520–600 nm filters, respectively.

**Genotyping**. Leaves were collected in 2 mL tubes containing three 2 mm glass beads and homogenized using TissueLyser II (Qiagen) at 30 Hz for 1 min twice. A simple extraction buffer (1 M KCl, 0.1 M Tris-HCl buffer pH 7.5, 10 mM EDTA) was added and boiled at 98 °C for 10 min. Tubes were centrifuged for 5 min at 11,000 rcf and the supernatant was collected in a new 1.5 mL tube and mixed with equal volume of 100% isopropanol. After brief vortexing, the tubes were centrifuged at 11,000 rcf for 15 min. The pellet was washed with 70%. (v/v with water) ethanol and dried at RT and respended in sterilized MilliQ water prior to a PCR reaction with primers listed in Supplementary Table 3.

**D14L CRISPR constructs and characterisation**. To generate Clustered Regularly Interspaced Short Palindromic Repeats (CRISPR) alleles two single guide RNA (sgRNA)s targeting exons of D14L were designed using the program, CRISPR-P (http://crispr.hzau.edu.cn/CRISPR/, version 1.0)[83]. The sgRNAs were comprised of $(N)_{20}GG$ where NGG served as Protospacer Adjacent Motif (PAM, N is any nucleotide) and ones with the least off-targets were chosen. The sgRNAs were cloned into the Gateway compatible vector system designed for rice[84]. Briefly, a pair of oligonucleotides was synthesized for each small guide RNA (Supplementary Table 3) to create overhangs to be ligated to the entry vector that was digested with the restriction enzyme, BsaI. Finally, Gateway LR reaction (Invitrogen) was performed to incorporate sgRNAs into the destination vector. Plant transformation was performed in wild-type cultivar Nipponbare by selecting hygromycin resistant calli[59]. Gene editing of primary transformants was verified by sequencing PCR products (Supplementary Table 3) around the PAM sequence. In the subsequent generation, plants carrying homozygous mutations but not the *CAS9* construct (Supplementary Table 3) were chosen for seed propagation. Finally, the T3 generation of *d14l* CRISPR alleles was used for the AM colonisation assay and gene expression studies.

**SMAX1 complementation line**. The *SMAX1* coding region was synthesized by GeneArt (Life Technologie) and the *SMAX1* terminator (800 bp after stop codon) was PCR amplified using primers listed in Supplementary Table 3. The coding sequence was cloned between rice *Actin* promoter and *SMAX1* terminator according to GoldenGate Cloning system to create an expression cassette flanked by Gateway attL1 and L2 sites, which was then recombined into the binary destination vector pEW343-R1R2 containing a hygromycin selection cassette[81,82].

**RNA-seq sample preparation**. Rice seeds were grown in 0.6% Bacto agar plates for 7 days at 30 °C in the dark. Seedlings were transferred to sand and grown for two weeks without any fertilizer supplementation in the same growth chamber described above. We collected three replicates comprised of eight plants per replicate for each genotype to minimize the cultivar effect. Each biological replicate was moved into a 60 mL tube containing 30 mL half- strength Hoagland solution (Phosphate 25 µM) for four days before harvest. Total RNA was extracted using Trizol followed by $LiCl_4$ precipitation. DNA was removed from 4 µg of RNA by Turbo DNase treatment followed by Column purification. DNase free RNA was confirmed by PCR and RNA quality was assessed by Bioanalyzer RNA6000 pico chip (Agilent). The RNA-seq library was generated with 2 µg of total RNA using TruSeq Stranded mRNA Library Prep Kit High Throughput (Illumina, RS-122-2013) according to the manufacturer's protocol (Catalog # RS-122-9004DOC, Part # 15031047 Rev.E). The library quantity and quality were measured using DNA1000 chip on Bioanalyzer 1200 (Agilent Technology). In total 18 samples (six genotypes and three biological replicates) were multiplexed and sequenced on a NextSeq500 as a $2 \times 75$ nt paired-end run.

**RNA-seq data analysis**. Raw fastq reads were cleaned by removing the adapter and the low-quality sequences using Trimmomatic v0.38[85]. Mapping of the cleaned reads was performed by TopHat2 with the default settings against MSU7 version of the rice reference genome (http://rice.plantbiology.msu.edu)[86]. The resulting BAM files were subjected to Cufflinks for transcript assembly and then to Cuffquant and Cuffnorm pipeline[87] to quantify the transcript levels. Transcript assembly was guided by the reference MSU7 annotation (http://rice.plantbiology.msu.edu/pub/data/Eukaryotic_Projects/o_sativa/annotatioa_dbs/pseudomolecules/version_7.0/

all.dir/) and all the parameters used in Cufflinks suite were set to the default. Integrative Genomics Viewer (IGV)[88] was used to visualise sequencing data.

**Preprocessing RNA-seq data and differential gene expression.** To address for outliers in sequencing replicates, non-metric multidimensional scaling was performed. Replicate 1 of Shiokari sample was identified as an outlier and was excluded from the subsequent analysis. Preprocessing and filtering of the RNA-seq data was done by R package edgeR[89], leaving 21,866 genes to be analysed in downstream steps. Differentially expressed genes (DEGs) were identified by using the R package DESeq2[90] for all 15 pairwise comparisons among 6 genotypes with the threshold of fold-change >2 and adjusted $p$-value < 0.05. It should be noted that different cultivars could affect the phenotypic characterization and the gene expression analysis due to the heterogeneous genetic make-up of each individual plant derived from a mix of traits segregating in the background. The heatmaps for DEGs from all pairwise comparisons were plotted by row standardizing the expression values to z-scores. A Z-score shows a single gene expression value relative to the mean of that of all six genotypes. A subset of DEGs were validated by qRT-PCR with cDNA from the same RNA and confirmed in a second independent experiment.

**Clustering differentially expressed genes.** To obtain smax1-UP and -DOWN, three DEGs from the comparisons, smax1 vs Dongjin wild-type, d3/smax1 vs Dongjin wild-type and d3/smax1 vs Shiokari wild-type, were combined. Second, two DEGs from the comparisons, d3/smax1 vs d3 and d3/smax1 vs d3/SMAX1 were combined. These were then clustered based on their expression patterns across six genotypes by fuzzy c-means clustering by applying the R package Mfuzz[44]. Median FPKM (Fragments Per Kilobase of exon model per Million reads mapped) values of replicates for each genotype were used to reduce complexity. During this process, 3 genes (LOC_Os01g29530, LOC_Os06g37760, and LOC_Os11g30690) were lost because the median value across all samples was 0. To determine the number of clusters $k$, the range of feasible $k$ was initially selected by observing where the minimum centroid distance among the clusters for each $k$ starts to plateau. From the range, $k$ was pinpointed by observing the significance of Gene Ontology term enrichments for each cluster. From these clusters, several clusters were chosen based on their gene expression patterns across six genotypes. We chose the patterns showing consistent up and downregulation in smax1- and d3/smax1- compared to the rest of the genotypes. Finally, we performed manual curation based on pattern supported by adjusted $p$-value < 0.05 to finalize the list.

**Gene ontology enrichment analysis.** To generate a more comprehensive list of rice genes with Gene Ontology (GO)[91] terms, Arabidopsis GO terms from the NCBI gene database[92] were transferred by orthology and integrated with rice japonica GO terms from Ensembl plant Biomart[93]. We used reciprocal BLASTp (version 2.7.1+) with the e-value threshold of 1E-10 to identify strict orthologues of rice and Arabidopsis. Only the terms annotated in *Arabidopsis thaliana* with experimental (EXP, IDA, IPI, IMP, IGI, IEP) and traceable curated (TAS) evidence were transferred. Gene set analysis was performed with one-tailed Fisher's exact test and the $p$-values were corrected by simulations which the component genes were randomly selected 1000 times. Only the GO terms of biological process class with more than three genes were analysed. The threshold for enrichment was the simulation corrected $p$-value of 0.05.

**Root exudate collection and strigolactone isolation.** Seeds were surface sterilized with 2% bleach for 30 min and germinated on 0.6% Bacto agar at 30 °C in darkness for 1 week. Six seedlings were transferred to 1.5 litre pots filled with autoclaved sand (five replicates of six seedlings per genotype). The plants were grown for 3 weeks under a 16 hour photoperiod ($150\ \mu M.m^{-2}s^{-1}$) in a temperature-controlled greenhouse at 28 °C with 65% relative humidity. Each pot was supplied with demineralised water for two weeks, and then with half-strength modified Hoagland's nutrient solution containing 10% phosphorus (P; 40 μM) for 1 week. Root exudates were collected from 3-week-old rice plants by adding and draining 300 mL water (containing 5% ethanol) through each pot. The exudates were frozen using liquid nitrogen and stored at −80 °C until used. For strigolactone isolation, 50 mL of root exudates were passed through preconditioned DSC-18 SPE cartridges (500 mg per 6 mL). After sample application, the cartridges were washed with 12 mL water (two times, 6 mL), and the strigolactones were subsequently eluted with 6 mL of 100% acetone (two times, 3 mL). The extracts were evaporated using SpeedVac, and the residues were redissolved in 50 μL of ethyl acetate. GR24 (5 pmol) was added to each sample as internal standard. For further purification, 4 mL of hexane was added to the solution and was loaded on preconditioned Phenomenex Strata SL-1 Silica (200 mg/3 mL) columns. The columns were washed with 2 mL of 100% hexane, and SLs were eluted with 3 mL of ethyle acetate: hexane (9:1). The solvent was evaporated in a SpeedVac, and the residues were redissolved in 100 μL of acetonitrile: water (25:75, vol/vol). The solutions were filtered through Micro-Centrifuge 0.2 μm NYLON filters and stored at −20 °C until analysis[50].

**Strigolactone analysis using LC-MS.** Strigolactones were analyzed using an Acquity UPLC™ System (Waters, Milford, MA, USA) coupled to a Xevo® TQ-XS tandem quadrupole mass spectrometer (Waters MS Technologies, Manchester,

UK) with an electrospray interface. The separation was performed on an Acquity BEH C18 column (2.1 × 100 mm, 1.7 μm) at flow rate 0.45 mL/min and temperature 45 °C, with 12 min binary gradient elution as follows: 0–0.4 min (85% A), 0.4–5 min (60% A), 5–8 min (35% A), 8–8.7 min (35% A), 8.7–9.5 (5% A), column wash for 0.8 min (5% A) and final column equilibration for 3 min for initial conditions (85% A), where A = 15 mM formic acid/water and B = 15 mM formic acid/acetonitrile (v/v). The eluate was introduced in the ion source of the mass spectrometer and analysed using the following settings: ion source/desolvation temperature (120/550 °C), desolvation/cone gas (nitrogen) flow (1000/150 L/hr), collision gas (argon) flow 0.15 ml/min. Compounds were analyzed in multiple reaction monitoring mode (MRM) and quantified by diagnostic transitions of: 4-deoxyorobanchol (331 > 97), putative methoxy-5-deoxystrigol isomers (361 > 97) using optimized cone voltage (20–22 V) and collision energy (20 eV). The MasslynxTM software, version 4.2 (Waters) was used to operate the instrument, acquire and process the data.

**Accession number.** SMAX1 (LOC_08g15230), D14L (LOC_Os03g32270), D3 (LOC_Os06g06050), Cyclophillin2 (LOC_Os02g02890), GAPDH (LOC_Os08g03290), Polyubiquitin (LOC_Os06g46770), RiEF1a (ABB90955), AM1 (LOC_Os04g04750), AM3 (LOC_Os01g57400), AM14 (ARK1, LOC_Os11g26140), PT11 (LOC_Os01g46860), DLK2a (LOC_Os05g15240), NFR5 (LOC_Os03g13080), LYK1 (LOC_Os01g36550), KIN6 (LOC_Os04g39180), DXS2 (LOC_Os07g09190), PSY (LOC_Os12g43130), OR (LOC_Os02g33149), Z-ISO (LOC_Os12g21710), ZDS (LOC_Os07g10490), D27 (LOC_Os11g37650), NSP2 (LOC_Os03g15680), DIP1b (LOC_Os11g06180), CCD7 (LOC_Os04g46470), CCD8, (LOC_Os01g54270), MAX1 (CYP711A2, Os01g0700900, LOC_Os01g50530; CYP711A3, Os01g0701400, LOC_Os01g50580), Actin1 (LOC_Os03g50890).

## Data availability

The data presented in this article are available in Supplementary Information file. The raw data of transcriptomics were deposited on GEO under the ID PRJNA593368. Other biological materials are available upon request to the corresponding author. The individual data point underlying all the figures, additional images of representative microscopic images, and original gel/blot images are provided as a Source Data file.

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

## Acknowledgements

We thank Lesley Plücker, Floriana Misceo for assistance in characterising the *D14L* CRISPR lines, Junko Kyozuka for providing *d3-1, d14-1* and *d27-1* seeds, Pawel Baster for conducting the RNA-seq experiment, and Anne Bate for technical support. J.C. was supported by EMBO Long-term research fellowship (ALTF 117-2014, co-funded by the the European Commission EMBOCOFUND2012, GA-2012-600394 from Marie Curie Actions), the Leverhulme Early Career Fellowship (EFC 2016-392) and the Isaac Newton Trust, Cambridge, UK. J.C. T.L. G.O. and U.P were also supported by the Biotechnology and Biological Sciences Research Council as BB/P003419/1 and by the Bill & Melinda Gates Foundation as OPP1028264. Additionally, H.J.B. was supported by the European Research Council (ERC Advanced grant CHEMCOMRHIZO, 670211). S.B. and E.J.W. were supported by the Biotechnology and Biological Sciences Research Council BB/P003176/1.

## Author contributions

Conceptualisation, J.C., U.P.; Investigation, J.C., T.L., E.K.S., W.S., M.R.; Methodology, J.C., T.L., J.Cho., S.B., B.P., M.R., K.A., G.A., H.B., E.W., G.O.; Writing, J.C. U.P. Funding Acquisition, J.C., U.P., H.J.B., E.J.W. All authors have contributed to and approved the final manuscript.

## Competing interests

The authors declare no competing interests.
