## [Peer Review File · Nature Communications]

Reviewers' comments:

Reviewer #1 (Remarks to the Author):

The core genes involved in strigolactone and karrikin signaling have been identified through genetic studies in rice and Arabidopsis over the past 15 years. In this manuscript, Choi et al. follow up on the 2015 discovery that D14-LIKE (D14L), an ortholog of KAI2 in Arabidopsis, is required for arbuscular mycorrhizal (AM) symbiosis. This study develops novel alleles of D14L through CRISPR-Cas9 gene editing and very nicely confirms the results achieved previously with the more complex hebiba mutant, which has a large deletion. The putative targets of KAI2 in Arabidopsis are SMAX1 and SMXL2, and loss of these genes suppresses MAX2/D3-dependent phenotypes that are associated with KAI2 signaling. Here, Choi et al. investigated whether SMAX1 also acts downstream of D14L and D3 in rice to regulate the potential for AM symbiosis. Through genetic analysis they demonstrate that this is indeed the case. They show that *smax1* is able to suppress *d14l* and *d3* AM symbiosis defects and that SMAX1 influences the expression of many AM symbiosis-associated genes. Although at this point it is perhaps unsurprising that SMAX1 acts downstream of D14L and D3, these results are important and impactful to the field, as they underscore the significance of the karrikin pathway for AM symbiosis and provide new clues as to how that regulation occurs.

Except for cases noted below, the authors have used appropriate controls in their experiments and performed appropriate statistical analysis of their data.

Major points

1. I have some concern about the mixing of cultivars in the generation of the double mutant lines. This may complicate the interpretation of the results in some cases, particularly if the phenotype is subtle. This may affect some of the gene expression results, including the transcriptomic analysis. Differentially expressed genes were pulled out of all 15 possible comparisons among the six genotypes; two genotypes are from Dongjin background and two from Shiokari, with the remaining two an unknown and not necessarily equivalent admixture of the two cultivars. I would expect there to be DEGs identified simply by comparing the two wildtype cultivars; similarly, some of the mutant vs. double knockout comparisons will be the result of expression differences due to various polymorphisms between the cultivars and not the mutations of interest. Another example is the measurement of SLs (Supplementary Figure 15). Dongjin and Shiokari have quite different SL profiles. So when assessing the *d3 smax1* mutant, which is derived from a mix of both backgrounds, what is the appropriate control to compare against? This being said, I recognize the difficulty the authors were facing and I do not think it reasonable to ask for these experiments to be performed again in a single background (e.g. through more gene editing). Instead, it is important that they acknowledge directly in the manuscript this potential limitation to their study and how it may affect their conclusions.

2. I'm having difficulty finding the set of genes referred to in lines 241-247. Please point out which gene expression pattern category(s) you're referring to in Table 3e, or make it clearer in another way. I am doubtful of the statement in lines 243 and 244 that the 22 genes were mainly involved in SL biosynthesis as there aren't that many known SL biosynthesis genes.

3. I would advise the authors to be cautious about the claims of synergistic effects for *d3/smax1* (line 246) without further validation. The synergistic effect is possible, for example if SMAX1 and D53 have opposite effects on the same targets. But, depending on how many genes were identified as differentially expressed between *d3/smax1* and WT, it is also possible that some or many of these 240 DEGs are just false-positives that were allowed with the 5% FDR cutoff.

4. Although I am inclined to believe it, the authors do not have enough evidence to support their statement that they show rice SMAX1 is a target of the D14L/D3 receptor (line 90) or how SMAX1

removal occurs (line 367). The biochemical and protein-protein interaction data are lacking. Instead, they will need to discuss the literature on D53 and SMXL6/7/8 to explain why this is a likely conclusion and why their d3 and d14l protoplast data are consistent with that model.

5. I'm having difficulty understanding the z-score method for showing expression levels in the heat maps. Is the idea that genes with higher or lower expression compared to the mean expression level of all genes were assigned larger z-scores? If so, is this problematic for a gene that undergoes a several fold change in expression but has a middle-of-the road level of expression? Please explain in more detail.

Minor points

6. The authors do not directly address the presence of SMAX2 in rice. They should at least point out in the main text that there is another close paralog of SMAX1 that emerged after the divergence of monocots and dicots, which may share similar functions. Is there any expression data (e.g. enrichment in roots) that would implicate SMAX1 as a more likely candidate than SMAX2 for a role in AM symbiosis?

7. The extended discussion of gene expression results tends to get confusing in places (e.g. lines 318-329). I suggest to make it more concise wherever possible.

8. What tool was used to predict the NLS in SMAX1?

line 57 - add reference for fungal metabolism statement

line 79 - add references for karrikin binding statement

line 82 - Bythell-Douglas 2017 BMC Biology is a much more thorough analysis of this evolutionary history; I suggest adding the reference to it

line 85 - ClpB/Heat Shock Protein 101 is not part of the AtSMAX1 gene name; given the low similarity between the two proteins, nor is that necessarily the function of AtSMAX1

line 107 - missing a parenthesis

line 126 - this is a very C-terminal insertion that leaves most of the coding sequence intact; it might be worth adding a comment about what's still present (e.g. the EAR motif), although based on the qRT-PCR analysis it seems transcription of the mutant form of the gene may terminate prior to the T-DNA insertion point

line 153 - I would argue that the transgene-positive plants are still homozygous *smax1* mutants

line 281 - Why aren't CCD7 and CCD8 discussed here with the other SL biosynthesis genes? Add them.

line 370 - since these transient expression experiments were done in protoplasts, it does not seem accurate to describe this as "standard symbiosis assay conditions"

line 396 - Are the developmental phenotypes of *kai2* subtle? The literature shows a number of clear phenotypes.

lines 420-422 - This is a grand concluding statement, but I'm not seeing the evidence for it, particularly the last part.

many places (line 315, etc.) - What is meant by "spontaneously activated" transcription? To me it is confusing because it does not seem spontaneous. Rather it is the effect of a mutation. It would be clearer to just say "increased expression" or "upregulation," etc.

Reviewer #2 (Remarks to the Author):

The manuscript details an important role of SMAX1 in suppressing AM symbioses and provides evidence that SMAX1 acts downstream of D14L receptor and D3 F-box protein. This regulatory network is known to play important role in perception of an as yet unknown ligand (that can be

mimicked by karrikin) that influences seedling development and this paper extends this to an important role in AM. In addition, RNAseq analysis and strigolactone (SL) biosynthesis indicates activation of many AM genes in *smax1* and *smax1 d3* double in absence of AM, indicating a double negative system to enable activation of AM program that the authors propose is in response to presence of D14L ligand.

Major concerns

Data presented in the manuscript clearly shows that the AM phenotype of *d14l* and *d3* mutants is dependant on *SMAX1*. However, the data gathered on SL biosynthesis and the majority of the gene expression data (RNAseq, etc, including other key symbioses genes) only compares *d3*, *smax1*, *d3 smax1* double mutant and respective wild type lines. This data shows an important role for *SMAX1* downstream of *D3* (but as the authors point out the stronger effects of double mutant also suggests additional modes of action). However, this data on SL and gene expression does not enable the hypothesis that D14L (and potential ligand) influences these processes by acting through *SMAX1* to be tested. Therefore, the model shown in Fig 5 connecting D14L and ligand binding, *D3*, *SMAX1* and symbiotic genes and SL biosynthesis is still speculative. Indeed, there is little synthesis or comparison of data collected here to previous data collected on WT vs *d14l* mutant influence on SL and/or gene expression (Gutjahr et al 2015 Science). If the model shown in Fig 5 is true there should be good inverse correlation between genes that are misregulated in *d14l* and *smax1* (ie. gene that are up in *smax1* should be down in *d14l* and vice versa) but this is not able to be analysed in this data set and no attempt is made to integrate this data with previous data? This is even more curious considering that *D3* is also part of SL signalling pathway, while D14L appears to be specific to signalling pathway of as yet unknown ligand.

Related to this point, Gutjahr et al 2015 clearly showed that addition of SL could not restore phenotype of *d14l* mutant, suggesting that SL-deficiency was not cause of low AM phenotype. This should be mentioned in the manuscript as the model shown in Fig 5 implies this point.

The strigolactone biosynthesis pathway for rice shown in Fig 3e is incomplete. Additional information is required on expression of genes between carlactone and orobanchol. In particular, this pathway should include both of the SLs measured 4DO and MeO-5-DS-is1 (possibly in brackets next to orobanchol?) and present our current understanding of the different roles of the different *MAX1* rice homologs (see Yoneyama et al 2018 (<https://doi.org/10.1111/nph.15055>), which supersedes Zhang et al 2014). In rice, *in vitro* and *in vivo* feeding experiments have suggested roles for Os900, Os1400 and Os5100 in various steps of SL biosynthesis pathway in rice but Os1500 appears to be a pseudogene (see Yoneyama et al 2018). Therefore, please update present gene expression data to include Os1400 and Os5100 in various lines. In addition, for ease of comparison to other literature the naming conventions for rice *MAX1* homologs with roles in SL biosynthesis used in Yoneyama et al 2018 should be followed.

Minor

Please note there is still only limited evidence that D14L and *D3* interact *in vivo* and no direct evidence of formation of a D14L-*D3*-*SMAX1* complex in any species so far (for discussion see Machin et al 2019 <https://doi.org/10.1111/nph.16135>)- please make this clear.

Line 41 meaning not clear " We concluded that removal of *SMAX1*, resulting from D14L signalling activation, integrates de-repression of essential symbiotic programmes with strigolactone hormone production."

Fig S4C define ED and FF in legend

Interesting that elevated SMAX1 expression in *smax1-2* mutant did not suppress AM- is this because protein is non-functional? Also OE SMAX1 in *smax1-1* mutant did lead to small but sign decrease in IH and A (AM measures)- suggest more of SMAX1 may suppress AM (this also supported by small decrease in some AM marker genes (Suppl Fig 6)

Would be nice to have description of what parts of proteins were missing/disrupted in *smax1* mutants

Line 223- please state conditions roots were grown under for RNAseq experiments- non-mycorrhizal - I see this is in methods but would be useful in text, Fig 3 legend and description of suppl data.

Please present quantitative data for SL biosynthesis genes shown in Fig 3e in supplemental figure as it is not clear which changes were significantly different and there appears to be clear differences between *smax1* and *d3 smax1* double

Line 385 many conditions may lead to upregulation of SL levels, including but not limited to P scarcity. Therefore as no direct role for SMAX1 in P response was examined I suggest removing statements " It has been long known that many plant species increase SL biosynthesis in response to phosphate scarcity, conceivably to attract AM fungi. Roots lacking functional SMAX1 therefore recapitulate the high SL production seen under low phosphate conditions".

Line 421 " by integrating the multiple environmental cues to optimize the benefits of the symbiosis." not clear what environmental cues this sentence alludes to? The phrase "optimize the benefits" is also a little awkward and unclear- I suggest rephrasing concluding line to relate directly to this manuscript.

Reviewer #1 (Remarks to the Author):

The core genes involved in strigolactone and karrikin signaling have been identified through genetic studies in rice and Arabidopsis over the past 15 years. In this manuscript, Choi et al. follow up on the 2015 discovery that D14-LIKE (D14L), an ortholog of KAI2 in Arabidopsis, is required for arbuscular mycorrhizal (AM) symbiosis. This study develops novel alleles of D14L through CRISPR-Cas9 gene editing and very nicely confirms the results achieved previously with the more complex hebiba mutant, which has a large deletion. The putative targets of KAI2 in Arabidopsis are SMAX1 and SMXL2, and loss of these genes suppresses MAX2/D3-dependent phenotypes that are associated with KAI2 signaling. Here, Choi et al. investigated whether SMAX1 also acts downstream of D14L and D3 in rice to regulate the potential for AM symbiosis. Through genetic analysis they demonstrate that this is indeed the case. They show that *smax1* is able to suppress *d14l* and *d3* AM symbiosis defects and that SMAX1 influences the expression of many AM symbiosis-associated genes. Although at this point it is perhaps unsurprising that SMAX1 acts downstream of D14L and D3, these results are important and impactful to the field, as they underscore the significance of the karrikin pathway for AM symbiosis and provide new clues as to how that regulation occurs.

Except for cases noted below, the authors have used appropriate controls in their experiments and performed appropriate statistical analysis of their data.

We appreciate the recognition of the importance of this work.

Major points

1. I have some concern about the mixing of cultivars in the generation of the double mutant lines. This may complicate the interpretation of the results in some cases, particularly if the phenotype is subtle. This may affect some of the gene expression results, including the transcriptomic analysis. Differentially expressed genes were pulled out of all 15 possible comparisons among the six genotypes; two genotypes are from Dongjin background and two from Shiokari, with the remaining two an unknown and not necessarily equivalent admixture of the two cultivars. I would expect there to be DEGs identified simply by comparing the two wildtype cultivars; similarly, some of the mutant vs. double knockout comparisons will be the result of expression differences due to various polymorphisms between the cultivars and not the mutations of interest. Another example is the measurement of SLs (Supplementary Figure 15). Dongjin and Shiokari have quite different SL profiles. So when assessing the *d3 smax1* mutant, which is derived from a mix of both backgrounds, what is the appropriate control to compare against? This being said, I recognize the difficulty the authors were facing and I do not think it reasonable to ask for these experiments to be performed again in a single background (e.g. through more gene editing). Instead, it is important that they acknowledge directly in the manuscript this potential limitation to their study and how it may affect their conclusions.

*To reiterate, we included homozygous double mutant and *d3*/SMAX1 siblings, which corresponded to the segregating plants derived from the same crossing between *d3* (Shiokari) and *smax1* (Dongjin). We agree with the reviewer that different cultivars could affect the phenotypic characterization and the gene expression analysis due to the heterogeneous genetic make-up of each individual plant derived from a mix of traits segregating in the background. We therefore collected eight plants per replicate and three*

replicates in total per genotype to minimise the cultivar effect. We clarify this design now in the Material and Methods (Page 20, lines 517-518) but did not change the main text.

2. I'm having difficulty finding the set of genes referred to in lines 241-247. Please point out which gene expression pattern category(s) you're referring to in Table 3e, or make it clearer in another way. I am doubtful of the statement in lines 243 and 244 that the 22 genes were mainly involved in SL biosynthesis as there aren't that many known SL biosynthesis genes.

We thank the reviewer for spotting this oversight; Table 3e is now the last worksheet of Supplementary Data 3 (excel format), also containing a diagram with the expression pattern categories. Regarding the statement of "22 genes involved in SL biosynthesis", we corrected the sentence into "mainly involved in secondary metabolism including SL biosynthesis" (Page 10, line 235-236).

3. I would advise the authors to be cautious about the claims of synergistic effects for d3/smax1 (line 246) without further validation. The synergist effect is possible, for example if SMAX1 and D53 have opposite effects on the same targets. But, depending on how many genes were identified as differentially expressed between d3/smax1 and WT, it is also possible that some or many of these 240 DEGs are just false-positives that were allowed with the 5% FDR cutoff.

We thank the reviewer for the insightful comment. We have removed 'synergistic' but maintain emphasis on the observation that expression levels of many genes are higher in d3/smax1 double as compared to smax1 and d3 single mutants (Fig. 3e and Supplementary Fig.16). The text has been rephrased accordingly, now reading "for a considerable number of genes we noticed increased transcript abundance in d3/smax1 double as compared to d3 and smax1 single mutants" (Page 10, line 232-233; Page 15, line 385). Although the recovery of false-positives cannot be excluded, 5% FDR is considered a standard cutoff for the identification of differentially expressed genes and accordingly is widely applied to transcriptome analyses.

4. Although I am inclined to believe it, the authors do not have enough evidence to support their statement that they show rice SMAX1 is a target of the D14L/D3 receptor (line 90) or how SMAX1 removal occurs (line 367). The biochemical and protein-protein interaction data are lacking. Instead, they will need to discuss the literature on D53 and SMXL6/7/8 to explain why this is a likely conclusion and why their d3 and d14l protoplast data are consistent with that model.

We agree with the reviewer that although we can genetically place SMAX1 downstream of the D14L/D3 receptor complex, SMAX1 as a direct target of the receptor complex cannot be concluded in the absence of biochemical data. We have adjusted the text accordingly "Here we show that rice SMAX1 operates downstream of the D14L/D3 receptor" (Page 4, line 84).

As suggested by the referee, we also describe the D53, SMXL6/7/8 degradation mechanism in the results section on Page 8, Line 191-194.

5. I'm having difficulty understanding the z-score method for showing expression levels in the heat maps. Is the idea that genes with higher or lower expression compared to the mean expression level of all genes were assigned larger z-scores? If so, is this problematic for a gene that undergoes a several fold change in expression but has a middle-of-the road level of expression? Please explain in more detail.

A Z-score shows a single gene expression value relative to the mean of that of all six genotypes. If a Z-score is 0, it means that the value for that gene in this genotype is identical to the mean value of the six other genotypes. By contrast, the higher z-score means that the gene was highly expressed in that genotype compared to other genotypes. Therefore, direct comparison between one gene to another is not suitable after Z-score conversion. We understand that the legend on the figure might have caused the confusion so we have changed the legend to “row Z-score” from “Z-score” to clarify that the normalisation was applied to genotypes (rows) (Fig.3 and 4).

Minor points

6. The authors do not directly address the presence of SMAX2 in rice. They should at least point out in the main text that there is another close paralog of SMAX1 that emerged after the divergence of monocots and dicots, which may share similar functions. Is there any expression data (e.g. enrichment in roots) that would implicate SMAX1 as a more likely candidate than SMAX2 for a role in AM symbiosis? *As suggested by the reviewer we now included reference to the presence and expression of SMXL2 in rice (Page 15, line 398-403). On the basis of RNAseq data, SMXL2 expression levels are comparable to SMAX1 in untreated control roots. Although it cannot be excluded that SMXL2 also plays a role in AM symbiosis, the loss of function of SMAX1 alone was sufficient to fully recover the lack of AM colonization in both d14l and d3 mutants (Fig.2). This is consistent with limited functional redundancy of SMAX1 and SMXL2 for AM symbiosis.*

7. The extended discussion of gene expression results tends to get confusing in places (e.g. lines 318-329). I suggest to make it more concise wherever possible.
The text has been amended (Page 12, lines 303-311).

8. What tool was used to predict the NLS in SMAX1?
The potential NLS in SMAX1 corresponds to the functionally validated NLS of Arabidopsis SMXL7 (Liang et al., 2016, The Plant Cell). This information was included in the main text along with the reference (Page 5, line 106-107).

9. line 57 - add reference for fungal metabolism statement
We added the reference as suggested (Page 3, line 53).

10. line 79 - add references for karrikin binding statement
Considering a large volume of references, the most recent review was added in the reference (Page 4, line 74).

11. line 82 - Bythell-Douglas 2017 BMC Biology is a much more thorough analysis of this evolutionary history; I suggest adding the reference to it
The suggested reference was added (Page 4, line 76)

12. line 85 - ClpB/Heat Shock Protein 101 is not part of the AtSMAX1 gene name; given the low similarity between the two proteins, nor is that necessarily the function of AtSMAX1

We agree with the reviewer and have removed "ClpB/Heat Shock Protein 101" from the text.

13. line 107 - missing a parenthesis

A parenthesis is added

14. line 126 - this is a very C-terminal insertion that leaves most of the coding sequence intact; it might be worth adding a comment about what's still present (e.g. the EAR motif), although based on the qRT-PCR analysis it seems transcription of the mutant form of the gene may terminate prior to the T-DNA insertion point.

As the reviewer mentioned, the SMAX1 transcript is truncated between primer set CC' and DD' based on the RT-PCR data in Supplementary Fig. 4. However, whether this leads to the production of a truncated protein is unknown. If it does, the potentially truncated protein still contains most of the first P-loop NTPase I, but has lost the second P-loop NTPase including a potential EAR motif. Unfortunately, we do not have a SMAX1 antibody yet to detect the endogenous SMAX1 protein level. With this uncertainty in mind, we do not think it is helpful to dwell upon it in the text. Instead, we marked the potential transcript truncation site in Supplementary Fig.2.

15. line 153 - I would argue that the transgene-positive plants are still homozygous *smax1* mutants

*We are sorry but do not understand the referee's comment as the plants indeed have been genotyped for intentionally being homozygous for *smax1*, ectopically expressing SMAX1.*

16. line 281 - Why aren't CCD7 and CCD8 discussed here with the other SL biosynthesis genes? Add them.

*Both CCD7 and CCD8 were up-regulated in *smax1* mutant, which has been added to the text (page 11, line 268-269).*

17. line 370 - since these transient expression experiments were done in protoplasts, it does not seem accurate to describe this as "standard symbiosis assay conditions"

We agree with the reviewer and changed it into "without any fungal stimuli" (Page 14, line 354).

18. line 396 - Are the developmental phenotypes of *kai2* subtle? The literature shows a number of clear phenotypes.

*We apologise for the confusion. We meant to explain that the developmental phenotypes of *kai2* are less pronounced than the complete loss of AM colonization in the rice *d14l* mutant. However, we agree with the reviewer that the developmental phenotypes of *kai2* are clear. Thus, we removed the phrase "despite subtle" from the text (Page 15, line 377).*

19. lines 420-422 - This is a grand concluding statement, but I'm not seeing the evidence for it, particularly the last part.

We removed the last part of the concluding statement.

20. many places (line 315, etc.) - What is meant by "spontaneously activated" transcription? To me it is confusing because it does not seem spontaneous. Rather it is the effect of a mutation. It would be clearer to just say "increased expression" or "upregulation," etc.

We thank the reviewer for pointing out this potential for confusion. With "spontaneously activated" we wanted to refer to a triggering of gene expression in the absence of a symbiotic stimulus. We have removed "spontaneous" from the text to avoid confusion.

Reviewer #2 (Remarks to the Author):

The manuscript details an important role of SMAX1 in suppressing AM symbioses and provides evidence that SMAX1 acts downstream of D14L receptor and D3 F-box protein. This regulatory network is known to play important role in perception of an as yet unknown ligand (that can be mimicked by karrikin) that influences seedling development and this paper extends this to an important role in AM. In addition, RNAseq analysis and strigolactone (SL) biosynthesis indicates activation of many AM genes in *smax1* and *smax1 d3* double in absence of AM, indicating a double negative system to enable activation of AM program that the authors propose is in response to presence of D14L ligand.

Major concerns

Data presented in the manuscript clearly shows that the AM phenotype of *d14l* and *d3* mutants is dependant on SMAX1. However, the data gathered on SL biosynthesis and the majority of the gene expression data (RNAseq, etc, including other key symbioses genes) only compares *d3*, *smax1*, *d3 smax1* double mutant and respective wild type lines. This data shows an important role for SMAX1 downstream of D3 (but as the authors point out the stronger effects of double mutant also suggests additional modes of action). However, this data on SL and gene expression does not enable the hypothesis that D14L (and potential ligand) influences these processes by acting through SMAX1 to be tested. Therefore, the model shown in Fig 5 connecting D14L and ligand binding, D3, SMAX1 and symbiotic genes and SL biosynthesis is still speculative. Indeed, there is little synthesis or comparison of data collected here to previous data collected on WT vs *d14l* mutant influence on SL and/or gene expression (Gutjahr et al 2015 Science). If the model shown in Fig 5 is true there should be good inverse correlation between genes that are misregulated in *d14l* and *smax1* (ie. gene that are up in *smax1* should be down in *d14l* and vice versa) but this is not able to be analysed in this data set and no attempt is made to integrate this data with previous data? This is even more curious considering that D3 is also part of SL signalling pathway, while D14L appears to be specific to signalling pathway of as yet unknown ligand.

*We agree with the reviewer that it would be desirable to compare the published *d14l* with the new *smax1* expression profiles. However, the experimental design of Gutjahr et al. (2015) was considerably different using a single replicate for each of the 5 time points within 24h to monitor transcriptional changes in response to germinated fungal spore exudates. In contrast, three biological replicated and a single time point were used by Choi et al.. Therefore, the transcriptomes of *d14l* and *smax1* cannot easily be compared across the independent experiments and set ups.*

*However, to investigate whether the pattern of genes misregulated in *smax1* is inversely correlated with the ones in *d14l*, we performed qRT-PCR-based expression analysis on a set of representative genes. As expected we observed that *DLK2a* and *NFR5* are down-regulated in *d14l*. Of course genes whose expression level is low in the wild type control (e.g. *LYK1*), cannot be further suppressed in the *d14l* background. Additionally, we also examined SL biosynthesis gene expression patterns. While *DXS2* was at a constitutive level in *d14l*, both *D27* and *MAX1* (*Os01g0700900*) showed reduced transcript abundance in the mutant. This is contrasting to up-regulation of SL biosynthesis genes in *d3* where a negative feedback regulation is known to operate (e.g. Arite T. et al., 2007, Plant J). We believe that these additional qPCR results lend further support to our conclusion that the D14L signalling pathway regulates expression of genes involved in AM symbiosis and SL biosynthesis but does not exclude the existence of other signaling cues. The gene expression analysis of *d14l* is now discussed in the text (Page 10, lines 258-259; Page 11, lines 276-277) and added in the Supplementary Fig. 15.*

Related to this point, Gutjahr et al 2015 clearly showed that addition of SL could not restore phenotype of *d14l* mutant, suggesting that SL-deficiency was not cause of low AM phenotype. This should be mentioned in the manuscript as the model shown in Fig 5 implies this point.

*We would like to clarify that we did not intend to suggest that SL-deficiency accounted for the low AM phenotype and apologise for overlooking the mis-leading model (Figure 5). We had mentioned in the Discussion section of the submitted manuscript “The strong *d14l* symbiosis phenotype cannot easily be explained at present but might be attributable to the collective alteration of the transcriptional activity of several CSSP components.” and maintained this sentence (Page 14, lines 370-372)”. We also revised the model (Fig. 5) by placing SL biosynthesis gene activation parallel to symbiosis gene activation.*

The strigolactone biosynthesis pathway for rice shown in Fig 3e is incomplete. Additional information is required on expression of genes between carlactone and orobanchol. In particular, this pathway should include both of the SLs measured 4DO and MeO-5-DS-is1 (possibly in brackets next to orobanchol?) and present our current understanding of the different roles of the different MAX1 rice homologs (see Yoneyama et al 2018 (<https://doi.org/10.1111/nph.15055>), which supersedes Zhang et al 2014). In rice, in vitro and in vivo feeding experiments have suggested roles for Os900, Os1400 and Os5100 in various steps of SL biosynthesis pathway in rice but Os1500 appears to be a pseudogene (see Yoneyama et al 2018). Therefore, please update present gene expression data to include Os1400 and Os5100 in various lines. In addition, for ease of comparison to other literature the naming conventions for rice MAX1 homologs with roles in SL biosynthesis used in Yoneyama et al 2018 should be followed.

We thank the reviewer for the helpful suggestion. We revised the Fig.3e and added the conversion from carlactone, carlactonoic acid, 4DO to Orobanchol. We also included the relative expression patterns of two functionally characterized MAX1 genes across six genotypes in Fig.3e. As suggested, we used the CYP711 numbers along with full RAP-DB to match the standard naming conventions. Finally, we removed PDR15 as it lacks functional characterization.

Minor

1. Please note there is still only limited evidence that D14L and D3 interact in vivo and no direct evidence of formation of a D14L-D3-SMAX1 complex in any species so far (for discussion see Machin et al 2019

<https://doi.org/10.1111/nph.16135>)- please make this clear.

We agree with the reviewer and have rephrased the text accordingly (Page 4, Lines 84)

3. Line 41 meaning not clear " We concluded that removal of SMAX1, resulting from D14L signalling activation, integrates de-repression of essential symbiotic programmes with strigolactone hormone production."

We rephrased the sentence into "We concluded that removal of SMAX1, resulting from D14L signalling activation, de-suppresses essential symbiotic programmes and increases strigolactone hormone production." (Page 2, lines 38-40)

3. Fig S4C define ED and FF in legend

ED' and FF' correspond to primers now included in the description of Supplementary Fig.4c.

4. Interesting that elevated SMAX1 expression in smax1-2 mutant did not suppress AM- is this because protein is non-functional? Also OE SMAX1 in smax1-1 mutant did lead to small but sign decrease in IH and A (AM measures)- suggest more of SMAX1 may suppress AM (this also supported by small decrease in some AM marker genes (Suppl Fig 6).

We agree with the reviewer that it would be interesting to investigate SMAX1-mediated suppression of AM symbiosis development at greater depth, for instance to determine the correlation between the degree of suppression of AM symbiosis establishment and the level of SMAX1 abundance, or the relevance of spatio-temporal distribution of SMAX1. In our view however, this is beyond the scope of this manuscript.

NB: in smax1-2 the protein is likely functional as the T-DNA insertion occurred in the promoter region.

5. Would be nice to have description of what parts of proteins were missing/disrupted in smax1 mutants (same answer as to reviewer #1, point 14) *The SMAX1 transcript is truncated between primer set CC' and DD' based on the RT-PCR data in Supplementary Fig. 4. However, it is unknown whether the truncated transcript leads to the production of a truncated protein. If it does, the potential truncated protein still contains most of the first P-loop NTPase I, but lost the second P-loop NTPase including a potential EAR motif. Unfortunately, we do not have a SMAX1 antibody yet to detect the endogenous SMAX1 protein level. With this uncertainty in mind, we do not think it is helpful to describe it in the text. Instead, we marked the potential transcript truncation site in Supplementary Fig.2.*

6. Line 223- please state conditions roots were grown under for RNAseq experiments- non-mycorrhizal - I see this is in methods but would be useful in text, Fig 3 legend and description of suppl data. *We added the phrase, 'non-inoculated control' in the text (Page 9, line 215), the figure legend of Fig.3, and Supplementary Fig.13.*

7. Please present quantitative data for SL biosynthesis genes shown in Fig 3e in supplemental figure as it is not clear which changes were significantly different and there appears to be clear differences between *smax1* and *d3 smax1* double.

We provided the qRT-PCR data of SL biosynthesis genes in Supplementary Fig.16 (Page 11, line 266-273).

8. Line 385 many conditions may lead to upregulation of SL levels, including but not limited to P scarcity. Therefore as no direct role for SMAX1 in P response was examined I suggest removing statements " It has been long known that many plant species increase SL biosynthesis in response to phosphate scarcity, conceivably to attract AM fungi. Roots lacking functional SMAX1 therefore recapitulate the high SL production seen under low phosphate conditions".

We agree with the reviewer and removed the respective sentences.

9. Line 421 " by integrating the multiple environmental cues to optimize the benefits of the symbiosis." not clear what environmental cues this sentence alludes to? The phrase "optimize the benefits" is also a little awkward and unclear- I suggest rephrasing concluding line to relate directly to this manuscript.

(same answer as to reviewer #1, point 19) We removed the last part of the concluding statement.

Reviewers' comments:

Reviewer #1 (Remarks to the Author):

The authors have made a number of changes to the text in response to the prior critiques and have clarified several issues in their rebuttal, which I think would be useful to share with other readers for whom similar questions may arise. I have a few comments remaining.

1. In several places the authors note that SMAX1-GFP was observed in d14l and d3 protoplasts, but not wildtype protoplasts, after transient transformation. They interpret this as evidence that SMAX1-GFP is degraded because of the activity of an endogenous signal (e.g. line 386). This is potentially very interesting. However, this experiment lacks a control for equal transformation of SMAX1-GFP in all three protoplast genotypes. Can the authors address this problem?

2. I still have some concern about false-positives among the 240 DEGs (Supplementary Table 3e) that were uniquely found in d3/smax1 and not in d3 or smax1 alone. It would be good to see validation of a few of these unexpected transcriptional changes by RT-qPCR.

3. On re-reading, it is not clear how the authors have derived their idea that SMAX1 functions as a repressor. This seems to be assumed throughout the manuscript. Please clarify.

4. The authors should add their rebuttal statement "Different cultivars could affect the phenotypic characterization and the gene expression analysis due to the heterogeneous genetic make-up of each individual plant derived from a mix of traits segregating in the background." to the manuscript, either in the main section or at the least in the Methods where they note the pooling of plants in replicates. They have not adequately acknowledged this important potential weakness to their analysis.

5. It would also be useful for them to add their statement "A Z-score shows a single gene expression value relative to the mean of that of all six genotypes." to the Methods section for clarification.

6. line 146: Change "lower total colonisation relative to homozygous mutants" to "lower total colonisation relative to siblings lacking the transgene" or to "lower total colonisation relative to smax1-1"

7. line 196: Change "d14l, d3 protoplasts" to "d14l and d3 protoplasts"

8. I may have missed it, but there does not appear to be any reference in the text to Supplementary Figure 14b and c and their significance. The legend for Supplementary Figure 14c also needs more information to clarify what the samples shown in the graph are.

Reviewer #2 (Remarks to the Author):

My comments have been addressed except for one. Please state in text that , Gutjahr et al 2015 clearly showed that addition of SL could not restore phenotype of d14l mutant, suggesting that SL-deficiency was not cause of low AM phenotype. Please state unambiguously that this means other SMAX1 regulated processes in addition to SL must be required to regulate AM colonisation. I suggest rephrasing sentence to read "The strong d14l symbiosis phenotype cannot be explained by modification of SL production alone (Gutjahr et al 2015) but must be attributable to the collective alteration of the transcriptional activity of several CSSP components." (Page 14, lines 370-372)".

We thank the editor and both reviewers again for their time and the many helpful suggestions. Below, please find our point-for-point replies to their comments.

Reviewer #1 (Remarks to the Author):

The authors have made a number of changes to the text in response to the prior critiques and have clarified several issues in their rebuttal, which I think would be useful to share with other readers for whom similar questions may arise. I have a few comments remaining.

1. In several places the authors note that SMAX1-GFP was observed in d14l and d3 protoplasts, but not wildtype protoplasts, after transient transformation. They interpret this as evidence that SMAX1-GFP is degraded because of the activity of an endogenous signal (e.g. line 386). This is potentially very interesting. However, this experiment lacks a control for equal transformation of SMAX1-GFP in all three protoplast genotypes. Can the authors address this problem?

We agree with the reviewer that it is vital to exclude technical artifacts. May we however point the reviewer's attention to the fact that four not three genotypes were included in this experiment, referring to two different mutants plus the corresponding two different wild type cultivars. The transformation of protoplasts from the four genotypes was performed in parallel, using one plasmid preparation for the entire experiment.

In accordance with the reviewer's suggestion, we compared the abundance of transformed free GFP (see figure below). The comparable protein levels of GFP in both wild-type and d14l protoplasts confirmed that the genotype did not influence transfection efficiency.

We would however prefer to keep Figure 2e as it is since we consider the reproducibility of the genotype dependence of the absence vs presence pattern of SMAX1-GFP a strong argument to support our main conclusion, namely that SMAX1-GFP is stabilized in d14l and d3 mutants but below detection limit in Nipponbare and Shiokari wild types.

2. I still have some concern about false-positives among the 240 DEGs (Supplementary Table 3e) that were uniquely found in d3/smax1 and not in d3 or smax1 alone. It would be good to see validation of a few of these unexpected transcriptional changes by RT-qPCR.

We thank the reviewer for his/her persistence and agree that validation of gene expression data obtained through RNAseq is important. We therefore performed qPCR-based expression analysis on a set of representative genes. We included cDNA from two independently grown experiment. We selected a marker gene for D14L signaling (DLK2a) and four genes known to be induced during symbiosis (NFR5, LYK1, KIN6 and PT11). We re-arranged supplementary Figure 13 such to show the validation of the expression profiles in Supplementary Figure 13a and the induction of gene expression in response to root colonization in 13b. The RNAseq data indicated that DLK2a mRNA levels were significantly higher in smax1 and d3/smax1 double mutant which was confirmed by qPCR (Supplementary Figure 13a). Transcripts of the other four genes accumulated to higher levels in the double mutant only according to the RNAseq results. This was confirmed for the prime marker gene for arbusculated rice cells, PT11, which was indeed significantly upregulated in the d3/smax1 double mutant only (Supplementary Figure 13a). For the other three genes, however, gene induction was observed in both the smax1 and the d3/smax1 genotypes (Supplementary Figure 13a). As the unique upregulation in d3/smax1 was not consistently confirmed, we deleted the corresponding text from the manuscript (page 10).

3. On re-reading, it is not clear how the authors have derived their idea that SMAX1 functions as a repressor. This seems to be assumed throughout the manuscript. Please clarify.

The genetic characterisation of the epistatic relationship of rice SMAX1 and the D14L/D3 receptor unambiguously identifies SMAX1 as a suppressor of D14/D3. In addition, mutation of SMAX1 leads to increased fungal colonization, consistent with SMAX1 functioning as a suppressor of symbiosis development, downstream of D14L/D3. We completely agree with the reviewer and had no intention to imply that SMAX1 protein itself functions as a repressor, rather the opposite, we had hoped to clarify that "De-repression of genetic programmes in response to SMAX1 removal can include both induced or suppressed transcriptional activity." (page 10, lines 248-249). We have carefully read and edited the manuscript to avoid confusion (e.g. Discussion, page 14, line 354).

4. The authors should add their rebuttal statement "Different cultivars could affect the phenotypic characterization and the gene expression analysis due to the heterogeneous genetic make-up of each individual plant derived from a mix of traits segregating in the background." to the manuscript, either in the main section or at the least in the Methods where they note the pooling of plants in replicates. They have not adequately acknowledged this important potential weakness to their analysis.

The text has been amended as suggested (Page 22, lines 538- 541).

5. It would also be useful for them to add their statement "A Z-score shows a single gene expression value relative to the mean of that of all six genotypes." to the Methods section for clarification.

The text has been amended as suggested (Page 22, lines 542-543).

6. line 146: Change "lower total colonisation relative to homozygous mutants" to "lower total colonisation relative to siblings lacking the transgene" or to "lower total colonisation relative to smax1-1"

The text has been amended as suggested (Page 6, lines 144).

7. line 196: Change "d14l, d3 protoplasts" to "d14l and d3 protoplasts"

The text has been amended as suggested (Page 8, lines 193).

8. I may have missed it, but there does not appear to be any reference in the text to Supplementary Figure 14b and c and their significance. The legend for Supplementary Figure 14c also needs more information to clarify what the samples shown in the graph are.

We thank the reviewer and apologise for the confusion. The data shown in Supplementary Figure 14b and c show the induced expression of these marker genes upon mycorrhizal colonization of rice roots. We have now added these data to Supplementary Figure 13 as 'b and c', as it is here where we first mention these marker genes (page 10, line 250). Whereas Supplementary Figure 14 now focuses on the expression of these genes in the smax^{SMAX1} complemented line (referred to on page 10, line 254).

Reviewer #2 (Remarks to the Author):

My comments have been addressed except for one. Please state in text that, Gutjahr et al 2015 clearly showed that addition of SL could not restore phenotype of d14l mutant, suggesting that SL-deficiency was not cause of low AM phenotype. Please state unambiguously that this means other SMAX1 regulated processes in addition to SL must be required to regulate AM colonisation. I suggest rephrasing sentence to read "The strong d14l symbiosis phenotype cannot be explained by modification of SL production alone (Gutjahr et al 2015) but must be attributable to the collective alteration of the transcriptional activity of several CSSP components." (Page 14, lines 370-372)".

The text has been amended as suggested (Page 14, lines 36).